# Preconfigured cortico-thalamic neural dynamics constrain movement-associated thalamic activity

Perla González-Pereyra ®[1], Oswaldo Sánchez-Lobato[1], Mario G. Martínez-Montalvo[1], Diana I. Ortega-Romero ®[1], Claudia I. Pérez-Díaz ®[1], Hugo Merchant ®[1], Luis A. Tellez ®[2] & Pavel E. Rueda-Orozco ®[1] ✉

Neural preconfigured activity patterns (nPAPs), conceptualized as organized activity parcellated into groups of neurons, have been proposed as building blocks for cognitive and sensory processing. However, their existence and function in motor networks have been scarcely studied. Here, we explore the possibility that nPAPs are present in the motor thalamus (VL/VM) and their potential contribution to motor-related activity. To this end, we developed a preparation where VL/VM multiunitary activity could be robustly recorded in mouse behavior evoked by primary motor cortex (M1) optogenetic stimulation and forelimb movements. VL/VM-evoked activity was organized as rigid stereotypical activity patterns at the single and population levels. These activity patterns were unable to dynamically adapt to different temporal architectures of M1 stimulation. Moreover, they were experience-independent, present in virtually all animals, and pairs of neurons with high correlations during M1-stimulation also presented higher correlations during spontaneous activity, confirming their preconfigured nature. Finally, subpopulations expressing specific M1-evoked patterns also displayed specific movement-related patterns. Our data demonstrate that the behaviorally related identity of specific neural subpopulations is tightly linked to nPAPs.

Neural activity patterns at the single and population levels have been proposed to underlie sensorimotor and cognitive processes. For example, memory formation in hippocampal networks has been associated with place cell sequential organization that can be recapitulated in different behavioral states such as environment exploration[1,2], sleep[3], or even short oscillatory periods[1,4]. On the other hand, in cortical and subcortical sensorimotor networks, neural population dynamics in the form of sequential activation have been associated with different aspects of movement, such as movement initiation[5] or motor timing[6,7]. In turn, these complex population activity patterns appear to be composed of various activity patterns at the single neuron level. For example, in the rodent cortical and

subcortical areas, including input and output nuclei of the basal ganglia (BG), cutaneous whisker or forepaw stimulations induce diverse activity patterns at the single cell level, generally characterized by a short-latency response followed by a transitory inactivation period, and usually a second, long-latency response[8–12]. This organization has also been reported for different sensory modalities, such as the auditory or olfactory systems[13–15], and is present in both behaving and anesthetized animals, suggesting a preconfigured nature. Moreover, this stereotyped organization, sometimes referred to as "information packets", spans for hundreds of milliseconds after stimulation and has been proposed as a generalized code for cognitive and sensory processing[2]. More recently, these neural preconfigured activity

[1]Departamento de Neurobiología del Desarrollo y Neurofisiología, Instituto de Neurobiología, UNAM Campus Juriquilla, Querétaro, Mexico. [2]Departamento de Neurobiología Conductual y Cognitiva, Instituto de Neurobiología, UNAM Campus Juriquilla, Querétaro, Mexico. ✉e-mail: pavel.rueda@gmail.com

patterns (nPAPs) have been observed in the dorsolateral striatum in the context of motor timing[9], where the experimental bidirectional manipulation of these dynamics has led to underestimations and overestimations of specific time intervals[9,16].

While nPAPs have been linked to a variety of cognitive domains, including memory formation, sensory processing, and interval estimation, little is known about the implications of this apparently generalized code during movement production. In this context, previous reports have demonstrated that the motor cortical and thalamic areas (including the ventral lateral and ventral medial regions; VL/VM) are mutually indispensable to sustain movement-related activity[17]. On the other hand, previous observations have demonstrated that VL activity in primates and rodents efficiently predicts movement onsets hundreds of milliseconds in advance[17–19], and VM inactivation delays movement onsets[20]. Furthermore, rhythmic auditory and visual stimulation evokes rhythmic neural patterns at single cell level in the VL. These patterns can be used to timely predict movement onsets[21]. Finally, it has been proposed that a specific subgroup of pyramidal tract cortical neurons that project to the motor thalamus is implicated in movement execution, but not preparation[22], opening the possibility that VL/VM activity may also be segregated into different clusters related to specific movement parameters. Here we explored whether motor-related thalamic activity is constrained by preconfigured patterns of activity evoked by its main input, M1. First, we found that VL/VM activity evoked by general optogenetic M1 stimulation was organized into stereotypical activity patterns present in virtually all animals. These patterns were similar when slightly changing the temporal structure of the stimulation protocol, but most importantly, when comparing awake and anesthetized conditions, as well as naïve and highly trained animals in a forelimb movement task. Then, we analyzed whether these M1-evoked VL/VM activity patterns were associated with specific behaviorally related neural activity patterns. By recording the same neurons during M1 stimulation and task performance, we found that VL/VM subpopulations expressing specific M1-evoked patterns also expressed movement-related patterns. Finally, optogenetic activation of the inhibitory pathway from the substantia nigra pars reticulata (SNr) to VL/VM regions affected nPAPs and movement execution parameters. Our data demonstrate that the behaviorally related identity of particular neural subpopulations is tightly linked to nPAPs.

## Results

### Motor thalamus responses to M1 stimulation are organized as stereotypical activity patterns

We performed silicon probe-based multiunitary activity recordings in the motor thalamus of head-fixed behaving mice (Fig. 1a). In the first set of experiments, recordings were performed in animals that had been accustomed to the head-fixed conditions for 3 to 5 days; that is, no specific motor task was implemented. The recordings targeted the M1–VL/VM motor thalamus circuit. Specifically, we expressed and stimulated channelrhodopsin 2 (ChR2) in M1 and recorded neural dynamics in VL/VM regions (Fig. 1b, c). This general manipulation affected multiple groups of M1 projecting neurons, including intratelencephalic (IT) and cortico-cortical (CC) neurons not projecting to the thalamus, as well as pyramidal tract (PT) and cortico-thalamic (CT) neurons, which project to the VL/VM[23]. Based on previous evidence from cortical and subcortical networks[9,11,24], we first explored whether VL/VM activity would reflect passive cortical stimulation (i.e., non-associated with specific behaviors or events) as repeatable neural dynamics at the single and population levels. To this end, we optogenetically stimulated M1 with two five-stimulus trains with different inter-stimulus intervals (ISI; 300 ms, 3.3 Hz and 500 ms, 2 Hz). We chose the 300/500 ms ISI based on previous reports indicating this frequency as a proxy of the speed of forelimb movements in rodents when walking or trotting[9,25,26]. In this set of experiments, we recorded a total of 1018 neurons from 18 awake mice (1–3 recording sessions per animal). From these neurons, 641 presented stable responses to at least one stimulation protocol, and from those, 377 presented stable responses to both stimulation protocols (see methods). The stimulation protocols produced a variety of activation patterns that were easily visible in the raster plots (Fig. 1d). Neurons as a population presented baseline firing rates that were consistent with previous findings[27] (median 3 Hz, 25th and 75th percentiles, 1.3 to 5.5 Hz; Fig. 1e). A general visual examination of neural activity suggested the M1-evoked responses were characterized by short-latency transient activation, followed by a brief period of inactivation and a long-latency rebound (Fig. 1f). Most neurons seemed to maintain their response pattern in both stimulation protocols (300 and 500 ms ISI), but to formally explore a potential adaptation to the ISI, we first quantified the response latency for the three main components of the response. Histograms confirmed the presence of two groups of latencies for the incremental responses and one group for the decremental responses at the population level and for both ISI (Fig. 1g, h). When comparing these latencies between ISI, we found that short-latency (increase and decrease) and long-latency responses were similar. The median short-latency response was about 10 ms for ISI, suggesting that this is the direct M1-VL/VM connection (Fig. 1i). For their part, the decrease and rebound responses may be related to intrinsic properties of the cells or micro/macro circuit mechanisms. Then we analyzed if the magnitude of the M1-evoked responses could experience short-term adaptation to the different ISI[28]. First, the average response of all neurons confirmed that the prototypical increase-decrease-increase pattern in spiking activity was maintained for each stimulus of the train and in both ISI (Fig. 1j). However, when we compared the amplitude of the short-latency increase, we observed that even when both conditions presented similar average trends, the second to fifth stimuli of the train were significantly larger than the first stimulus only in the 300 ms ISI condition (Fig. 1k). This finding indicates that there may be a mechanism for short-term facilitation depending on the frequency of stimulation in the M1-VL/VM connection. The amplitude of the decremental response was not significantly different between stimuli or ISI conditions (Fig. 1l). Finally, for both ISI, the last two stimuli of the train produced significantly smaller amplitudes when compared with the first or second stimuli (Fig. 1m), indicating similar short-term adaptive mechanisms for the second excitatory component of the response for both conditions.

Then we explored whether VL/VM neurons as a population may dynamically adapt to two different ISI. First, we plotted the averaged (z-scored) activity of each of the 377 thalamic neurons that were active in both stimulation protocols aligned to the first cortical stimulus of the train. Neurons in both conditions were sorted according to the time of their highest firing rate between the first and second stimuli of the train in the 300 ms ISI protocol (300 and 500 ms; Fig. 2a). This representation suggested that, regardless of the ISI, the population activity organized into similar sequential activations that spanned around 300 ms after stimulus onset, even for the 500 ms ISI. This was confirmed when we plotted all active neurons in each protocol (i.e., not only neurons firing in both protocols) and sorted the activity of both conditions according to their response latencies to the 300 ms or 500 ms ISI protocol (Supplementary Fig. 1a). The same kind of sequential responses were observed in a subset of experiments (3 animals, 153 neurons) where the same stimulation protocols were applied at different light intensities ranging from ~0.6 to 20 mW (Supplementary Fig. 2, see methods). Then, upon aligning the average population response to the onset of each stimulus of the train, we identified virtually identical responses for both ISI (Fig. 2b). Next, the evoked population averages were compared with population responses constructed with spike trains randomly circularized from the activity of the same neurons (see methods; Fig. 2b, gray traces). Finally, for each neuron we calculated the correlation coefficient between the

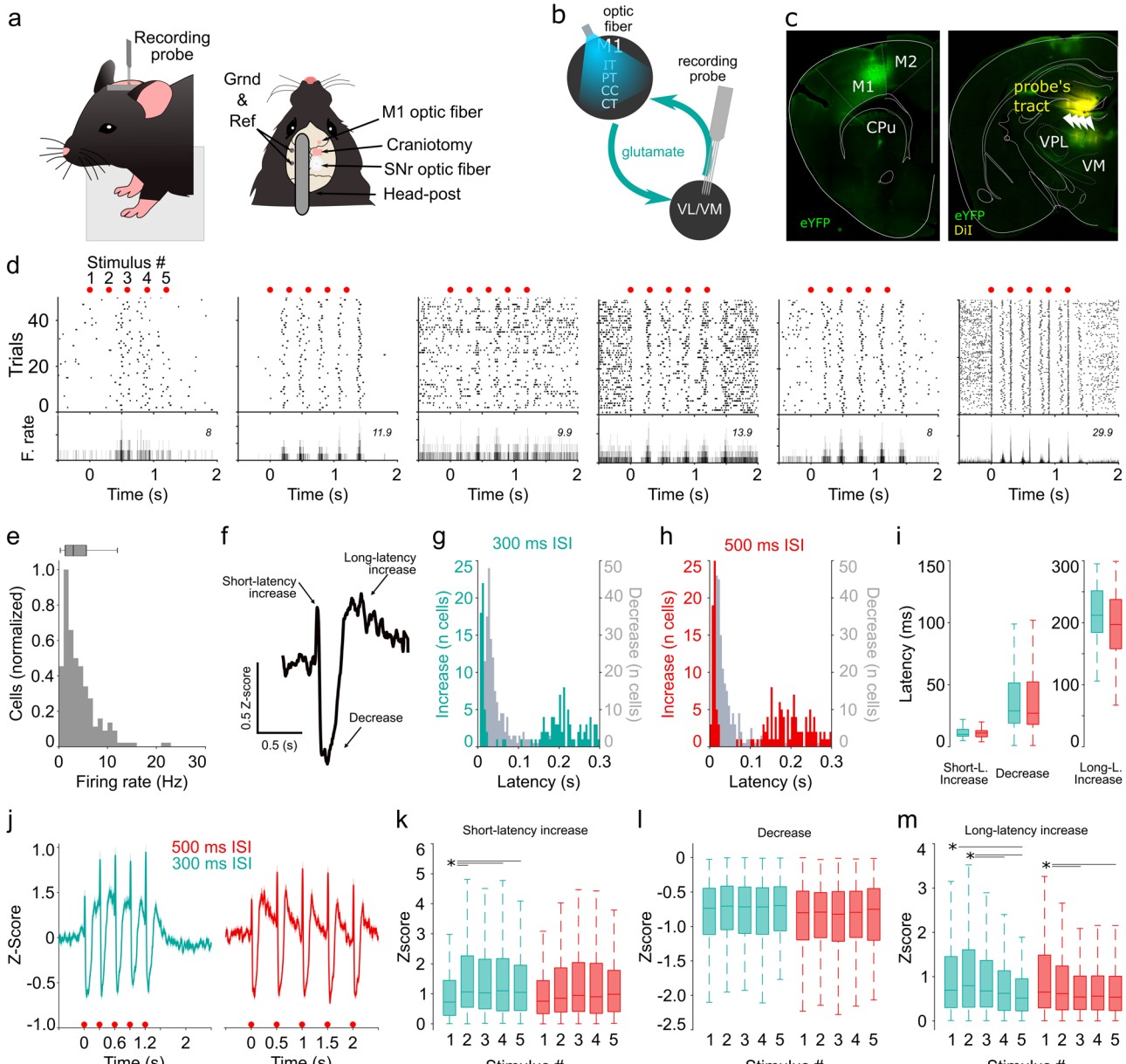

**Fig. 1 | Prototypical M1-evoked responses in VL/VM. a** Experimental set-up. **b** M1-evoked neural activity was recorded in VL/VM. **c** Representative histological confirmation of stimulation and recording sites in M1 and VL/VM. Probe shanks (DiI stained) are indicated with arrowheads. **d** Representative spike rasters and average peri-event histograms for neurons recorded in VL/VM, aligned to the first stimulus of the train (3.3 Hz; red dots). Maximum firing rates are indicated for each neuron. **e** Firing rate histogram for all cells. **f** Average population response for the first stimulus of the train. Response latency for the short-latency excitatory (color coded) and inhibitory (gray) components of the stereotypical response for the 300 (**g**) and 500 ms (**h**) ISI. **i** Response latencies in the 300 ms and 500 ms conditions. **j** Full train-averaged population response for the 300 ms and 500 ms conditions. Amplitudes for the short-latency (**k**), decremental (**l**), and long-latency (**m**) components of the train-evoked responses. Statistical differences (**i**, **k–m**) are indicated by asterisks and lines and were performed in 377 neurons that maintained stable conditions in both protocols. The central line and box in boxplots represent the median and 25th–75th percentiles, and whiskers extend to the most extreme datapoints excluding outliers. Kruskal-Wallis non-parametric one-way ANOVA (K-W) and Bonferroni post hoc test was applied in (**i**) (degrees of freedom $[df] = 5$; $X^2 = 483.83$; $p < 0.001$), k (300 ms, df = 4; $X^2 = 14.89$; $p < 0.005$; 1 vs 2 $p = 0.005$; 1 vs 3 $p = 0.08$; 1 vs 3 $p = 0.017$; 1 vs 5 $p = 0.04$; 500 ms df = 4; $X^2 = 5.86$; $p = 0.2$), l (300 ms df = 4; $X^2 = 3.42$; $p = 0.49$; 500 ms df = 4; $X^2 = 1.31$; $p = 0.859$) and m (300 ms, df = 4; $X^2 = 24.86$; $p < 0.001$; 1 vs 2 $p = 0.001$; 2 vs 4 $p = 0.04$; 2 vs 5 $p < 0.001$; 500 ms df = 4; $X^2 = 12.6$; $p = 0.013$; 1 vs 3 $p = 0.04$; 1 vs 5 $p = 0.03$).

activity evoked by each stimulus of the the 300 and 500 ms ISI trains and between the surrogated spiking activity. The correlation distributions between the 300 and 500 ms ISI trains were significantly higher than those comparing the surrogated spike trains (Fig. 2c, bottom right, gray bars). These data suggest that, as a population, the VL/VM network may not be temporally adapting to the two stimulating conditions. To further characterize this possibility, we applied a geometric approach[29,30] to compare the matrices from our two ISI

conditions. For the first comparison, called "relative" comparison, 30 bin matrices were constructed for both ISI; that is, the bin sizes were adjusted so that both ISI fitted 30 bins (10/16.6 ms bins for 300/500 ISI; Supplementary Fig. 1b). For the second comparison, called "absolute" comparison, 30 bin matrices were constructed with the first 300 ms (30 bins) after stimulus onset for both ISI (Supplementary Fig. 1c). We averaged the activity of the five stimuli of the train to obtain an averaged neural sequence for each ISI (Supplementary Fig. 1b, c). Then we

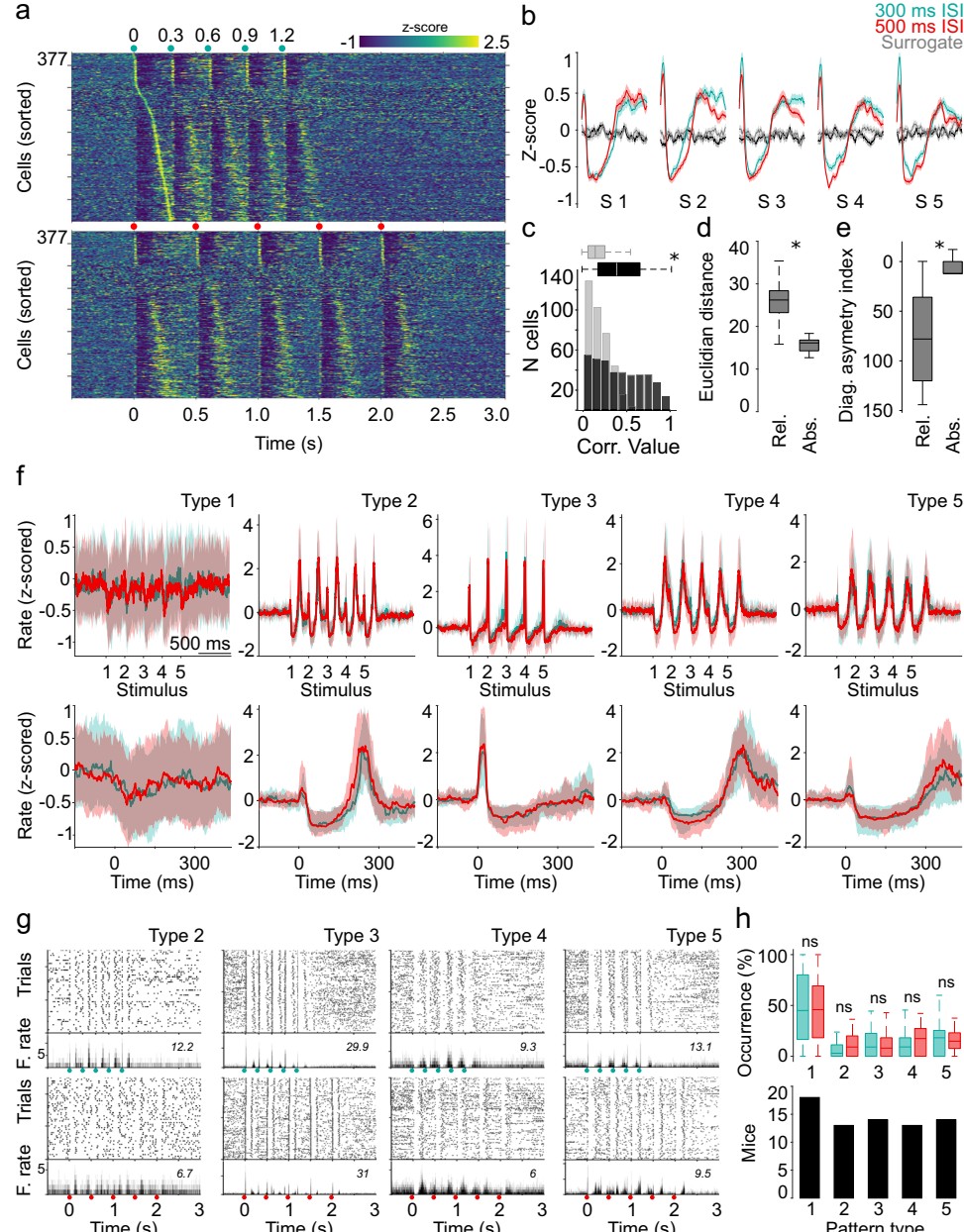

**Fig. 2 | VL/VM responses can be classified into distinct M1-evoked patterns.**
**a** Average firing rates (z-scored) evoked by M1 stimulation for 300 ms (top) and
500 ms (bottom) ISI protocols (green and red dots). Cells were time sorted
according to their highest firing rate between the first and second stimulus of the
train. **b** Averaged population activity for the 300 ms, 500 ms, and surrogate data
aligned to each stimulus of the train. Solid lines and shaded areas represent the
median and the 25th and 75th percentiles, respectively. **c** Correlation distribution
between each stimulus of the 300 ms and 500 ms trains and surrogate spiking
activity ($n = 377$ cells x 5 stimuli; two-sided Mann–Whitney, *$p < 0.001$). Euclidean
distances (**d**) and diagonal asymmetry index (**e**) between the relative and absolute
300/500 ms comparisons. Significant differences (two-sided Mann–Whitney,
*$p < 0.001$) were calculated on observed and surrogated values from the 377 neu-
rons that maintained stable conditions in 300 ms and 500 ms conditions. **f** Average
M1-evoked patterns for cells classified as specific pattern clusters for the 300 ms

and 500 ms conditions. Top and bottom rows display the full train and first sti-
mulus magnification evoked patterns, respectively. Solid lines and shaded areas
represent the median and the 25th and 75th percentiles, respectively.
**g** Representative spike rasters and average peri-event histograms for neurons
(same neuron in each column) belonging to specific pattern clusters recorded in
the 300 ms (top row) and 500 ms (bottom row) conditions. Activity was aligned to
the first stimulus of the train (green and red dots for 300 and 500 ms ISI, respec-
tively). Maximum firing rates are indicated for each neuron. **h** Percentage of cells
belonging to each pattern cluster displayed in (**f**) (upper panel, $n = 377$ neurons).
Number of animals that presented each pattern (lower panel). K-W non-parametric
one-way ANOVA, df = 9; $X^2 = 65.66$; $p < 0.001$. The central line and box in boxplots
(**c**–**e**, **h**) represent the median and 25th–75th percentiles, and whiskers extend to
the most extreme datapoints excluding outliers.

calculated Euclidean distance matrices for each ISI and across ISI for
relative and absolute comparisons (Supplementary Fig. 1d) and
reported the average Euclidean distance (Fig. 2d) and diagonal asym-
metry index (see methods; Fig. 2e) between the absolute and relative
matrices depicted in Supplementary Fig. 1b, c. Relative comparisons

presented significantly higher Euclidean distances and a diagonal
asymmetry index, indicating that when comparing bin by bin in real
time (absolute comparison), both 300 ms and 500 ms ISI produced
similar population dynamics. Our results indicate that under these
passive stimulation conditions, VL/VM appears to produce non-

temporally adapting stereotypical population responses evoked by each cortical stimulus of the train, suggesting a robust preconfigured organization in cortico-thalamic communication.

Next, we explored if the sequential stereotypical population responses could be composed of subgroups of neurons expressing specific activity patterns. To this end, we applied a principal component analysis (PCA)/Silhouette-based method of classification[11] (see methods) to the peri-event histograms obtained from the individual raster plots (Fig. 1d) over the entire population of cells and for both conditions (300 and 500 ms ISI). We applied this method to the 377 neurons with stable responses in both protocols. Because the 300 ms ISI protocol and the first 300 ms of the 500 ms ISI protocol produced virtually identical average responses for each stimulus of the train (Fig. 2a–e & Supplementary Fig. 1), we focus our analysis on the first 300 ms after each stimulus for both 300 and 500 ms data. The projection with the highest Silhouette values indicated that cells could be more effectively categorized into five distinct types of activation patterns (pattern clusters) evoked by M1 stimulation (Fig. 2f) and that all patterns were present in both ISI. The first pattern (Type 1) exhibited very low or non-response amplitudes. This pattern captured about 45-50% of the entire population of neurons (Fig. 2f, top). The remaining four patterns consisted of short-latency transient activations followed by inhibitions, sharp activations, or inactivation followed by rebounds (Fig. 2f). The structure of these patterns was maintained in both ISI and visible in the average response pattern (Fig. 2d) or individual representative neurons (Fig. 2g). These patterns were almost homogenously distributed over the other ~50% of the population of neurons (Fig. 2h, top). We confirmed these findings by applying a different analytical approach based on a generalized linear model (GLM) analysis (see methods; Supplementary Fig. 3). Then we estimated if all response patterns were present in the 18 animals recorded under these conditions. We found pattern type 1 in all 18 animals, but, interestingly, we also identified pattern types 2 to 5 in most of the animals (Fig. 2h, bottom). More importantly, we found that 12 out of the 18 animals exhibited the five patterns, two out of 18 presented four patterns, and only four animals presented only one pattern. Altogether, these results indicate that passive stimulation of M1 in awake, naïve behaving mice produces robust stereotypical patterns of thalamic activation preserved across animals, further supporting a preconfigured organization. If this were the case, it would also make sense that this organization would exist under different brain-state conditions, such as during anesthesia. To explore this possibility, we used three animals that were recorded for two sessions under the previously described conditions (the animals used for light intensity experiments depicted in Supplementary Fig. 2). In the third session, animals were administered a 1 g/kg dose of the anesthetic urethane, which is known to preserve different sensory-evoked dynamics in rats[9,11,14]. We observed that in this subgroup of animals, M1 stimulation evoked almost indistinguishable averaged population dynamics (Supplementary Fig. 4a, b) and response patterns (Supplementary Fig. 4c–e) in awake conditions compared to anesthetized conditions. It is important to highlight that anesthetized recordings were performed after two sessions in which at least 800 stimulation protocols were provided. Still, the response patterns remained stable, strongly supporting the conclusion of a preconfigured response while raising the questions of its specific origin and its role during movement execution.

Regarding the origin of these patterns, cortico-thalamic projections to VL/VM arise from two neural subpopulations, PT and CT neurons[23]. To disentangle the contribution of PT and CT subpopulations to these potentially preconfigured responses, we implemented the following approaches. For PT neurons ($n = 5$ animals) we implemented an anatomically based strategy with two viruses (Fig. 3a) as the one reported before for IT neurons in rats[24]. Here, we first induced the expression Cre recombinase (Cre) in M1 cortico-thalamic projecting neurons (including PT and CT) by injecting a retrograde virus (AAV-

retrograde-pgk-Cre AAV) in the VL/VM. Considering that PT neurons (but not CT neurons) projects to both, to the VL/VM and the dorsolateral striatum (DLS), four weeks after the first injection we induced the Cre-dependent expression of ChR2 and GFP in PT neurons by performing a second injection of a retrograde virus (in the ipsilateral DLS to the first injection; pAAV-EF1a-doublefloxed-hChR2-eYFP; Fig. 3a). Four weeks after these injections, we observed unilateral rYFP expression in layer V of M1/M2 regions (Fig. 3b) and we were able to optically evoke PT activity in M1 (Supplementary Fig. 5a–e). Then, M1 stimulation with the 300 ms ISI protocol induced weak thalamic responses characterized mainly by discrete pauses in activity induced by each stimulus of the train (Fig. 3c). This manipulation did not induce strong short-latency increases or long-latency rebounds observed with the original strategy, suggesting a marginal contribution of PT subpopulation to the described population dynamics. To further confirm this possibility, we performed the same 300 ms ISI protocol but using 50 ms light pulses (as opposed to the 5 ms ones used in previous manipulations) and with maximum light power. This manipulation induced similar dynamics to the 5 ms based protocol but with slightly stronger pauses but importantly, the short- and long-latency increases were also almost absent (Supplementary Fig. 5f).

To investigate CT and PT projections, we retrogradely induced the expression of ChR2 in PT and CT (PT + CT) neurons by injecting a retrograde virus in VL/VM (pAAVretro-Syn-ChR2(H134R)-GFP; $n = 2$ animals), and four weeks after we performed the same stimulation protocol producing very similar population responses to those observed in the original unspecific strategy depicted in Fig. 2a (Fig. 3d–f). Then, to further isolate the CT component, in a new group of animals ($n = 3$) we attempted to express ChR2 in CT neurons and eliminate PT, IT and CC neurons in the ipsilateral M1 to the recorded VL/VM. To this aim we introduced the following strategy: in a first surgery we performed three injections, a retrograde virus in the VL/VM to express ChR2 and GFP, and a retrograde virus in the ipsilateral DLS and the contralateral M1 to induce the expression of Cre (same virus as the ones used in the previous experiments). In this way, both, PT and CT neurons would express ChR2, and PT, IT and CC (but not CT) neurons would express Cre in the ipsilateral side to the recorded VL/VM (Fig. 3g). Four weeks later, a single injection of the virus pAAV-flex-taCasp-TEVp in the ipsilateral M1 to the recordings site induces the Cre-dependent expression of caspase 3 in PT, CC and IT neurons ultimately resulting in their apoptosis (Fig. 3g). Four weeks after the last injection we observed ipsilateral expression of GFP and a decreased signal in layer V (Fig. 3h). Her again the M1 300 ms ISI stimulation protocol induced similar population responses to those observed in our previous experiment (PT + CT) and our original observations (Fig. 3i). Finally, we explored the existence of nPAPs evoked by specific subpopulations. For this analysis, we pooled all neurons recorded from all groups, that is, the general M1 stimulation group displayed in Fig. 2, and the PT, PT + CT and CT groups, and re-applied our PCA/Silhouette-based method to identify groups of neurons with similar response patterns and compared their occurrence proportions. We found that in the isolated PT group the patterns with short-latency and long-latency activation were significantly reduced, while both groups PT-CT and CT presented all five patterns of activation (Fig. 3j) with similar proportions than the original group (Fig. 3k), except that PT + CT group expressed a significantly higher proportion of pattern 4 than the original M1 group. Altogether, these data suggest that M1-evoked nPAPs are mainly related to CT projections with a marginal component of PT neurons.

## Motor thalamus responses associated with movement execution

Next, we explored whether the M1-evoked thalamic nPAPs described in the previous sections could be related to specific parameters of movement execution. With this objective, we established a behavioral

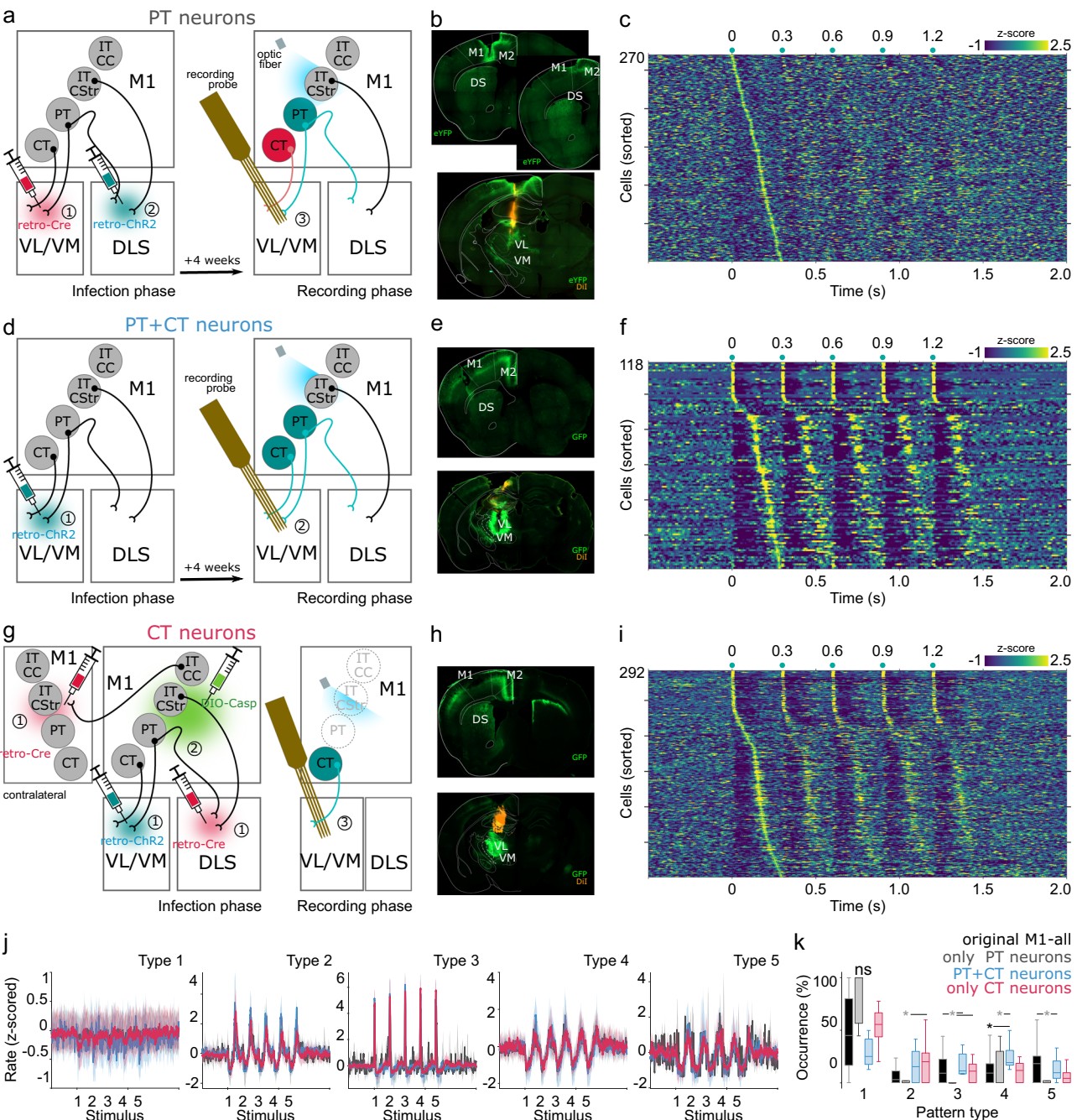

**Fig. 3 | PT and CT contribution to M1-evoked patterns.** Schematic representations (**a**, **d**, **g**) and representative histological confirmations (**b**, **e**, **h**) for infection strategies to specifically target PT (**a**, **b**), PT + CT (**d**, **e**) and CT (**g**, **h**) neurons (GFP and eYFP proteins were used as reporters). **c**, **f**, **i** Circled numbers inside the schemes indicate sequential order of injections and recordings. Average firing rates (z-scored) evoked by PT (**c**), PT + CT (**f**) and CT (**i**) M1 stimulation for cells recorded under 300 ms inter-stimulus intervals (ISI) protocol. Cells were sorted according to the moment of their highest firing rate between the first and second stimulus of the train. Each stimulus of the train is indicated above each panel (green dots). **j** Average M1 subpopulation-evoked patterns for cells classified as part of specific pattern clusters for the 300 ms ISI (color coded). Solid lines and shaded areas

represent the median and the 25th and 75th percentiles, respectively. **k** Percentage of cells belonging to each pattern cluster displayed in (**j**) (Boxplots indicate median and 75th and 25th percentiles, whiskers extend to the most extreme datapoints excluding outliers). Statistical comparisons were performed by applying K-W non-parametric one-way ANOVA and Bonferroni post hoc tests; df = 19; $X^2 = 145.94$; $p < 0.001$ (original M1 all, $n = 377$; only PT neurons, $n = 270$; PT + CT neurons, $n = 118$; only CT neurons, $n = 292$). Significant differences are indicated by asterisks and lines joining specific comparisons (Type 2, PT vs CT $p = 0.005$; Type3, Orig vs PT $p = 0.007$, PT vs PT + CT $p = 0.002$, PT vs CT $p = 0.012$; 0.04; Type 4 ctrl vs PT + CT $p = 0.04$; PT vs PT + CT $p < 0.001$; Type 5, Orig vs PT $p = 0.003$; PT vs PT + CT $p = 0.04$.

protocol compatible with our head-fixed recording setup. Mice ($n = 13$) were trained to handle a lever located under their left forepaw, contralateral to the recording sites (Fig. 4a). Behavioral protocols were divided into three phases (Fig. 4a, right). In the first phase, named "Handling and Modeling," the animals were accustomed to head-fixing

conditions for about 2–3 sessions (20–30 min per session, 1 session per day). Immediately after, animals were placed under water restriction and began a modeling period in which they were taught to displace the lever to receive rewards (water drops delivered through the waterspout). The modeling period lasted about 10 sessions

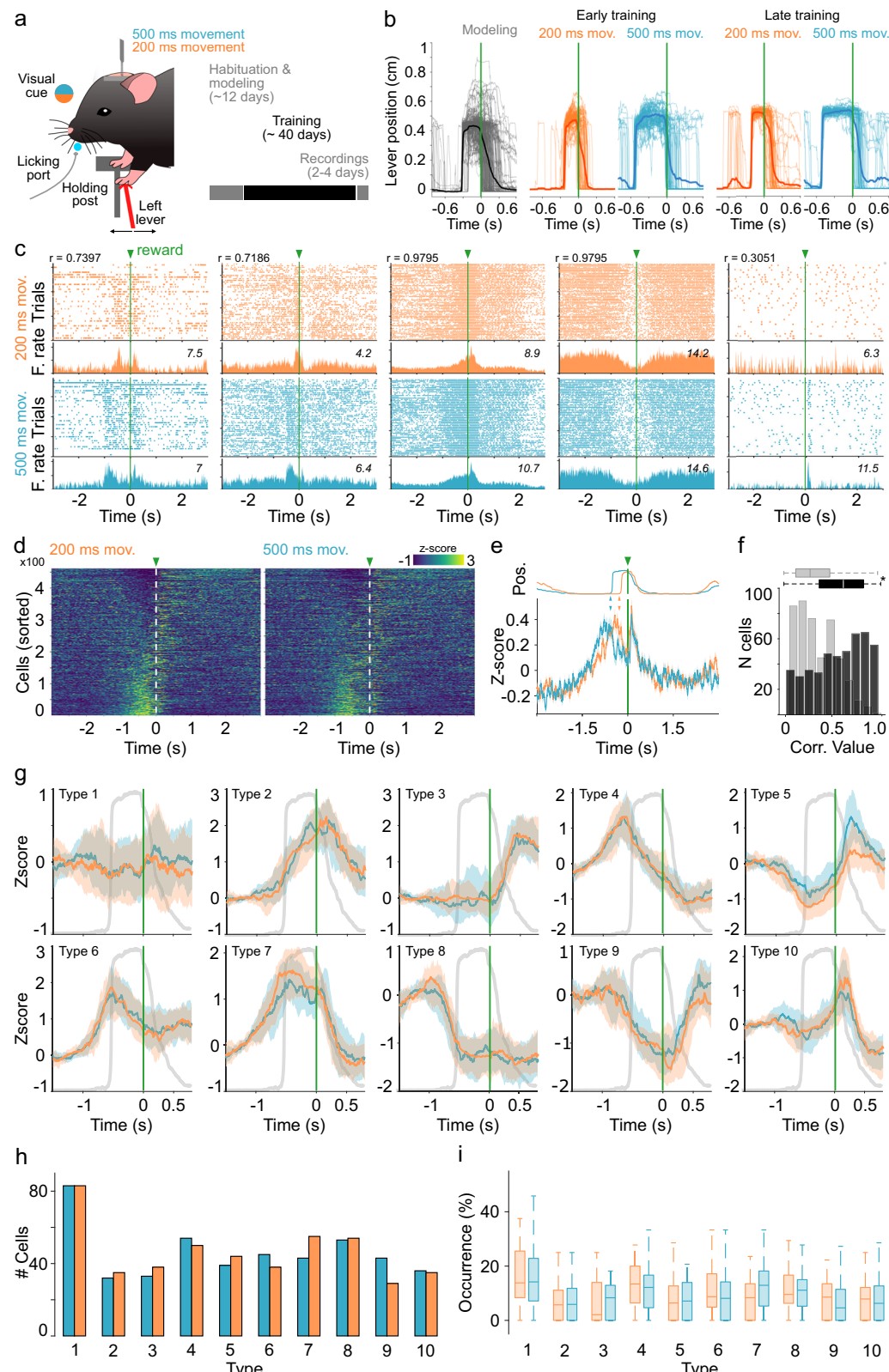

(50–60 min sessions, 1 session per day). A reward was delivered with progressively longer movements, starting from 50 ms (0.1 mm displacement threshold) to 500 ms displacements (0.8 mm displacement threshold). We rewarded movements in any possible direction, but individual animals quickly displayed a preference for either backward or forward directions (i.e., pushing or pulling displacements, respectively). Lever trajectories were constantly recorded, analyzed, and

aligned to reward delivery (Fig. 4b), indicated by a 1 s green light located in front of the mice. After the modeling period, animals started the formal training in the two-duration version of the task. Here mice were asked to perform 200 or 500 ms movements in alternating blocks of 20 consecutive trials for each duration. The 200 and 500 ms blocks were signaled by a contextual white or blue light, respectively. Movement duration was selected for two reasons: to match as closely

**Fig. 4 | Thalamic signals during movement execution. a** Behavioral setup.
**b** Representative trajectories for trials in the 200 ms and 500 ms blocks at different stages of learning. Lever trajectories are aligned to the reward onset (vertical green lines). **c** Pairs of spike rasters (top) and average peri-event histograms (bottom) for representative cells with different response patterns recorded during movement execution in the two-interval task during the 200 ms (top row, orange code) and 500 ms (bottom row, blue code) trials. Spiking activity was aligned to the reward onset (green lines and arrowheads). Maximum firing rates are indicated for each neuron. **d** Z-scored averaged firing rates for cells recorded during the 200 ms (left) and 500 ms trials time sorted according to the average maximum firing rate between −1.2 s and reward delivery (in zero, green arrow and withe dotted line). **e** Averaged peri-event histogram of the firing rates (bottom) for all cells recorded during the 200 ms and 500 ms trials aligned to the reward delivery (green line and arrow). The averaged lever trajectories for both types of trials are presented at the top of the plot. Color-coded arrows indicate movement onsets. **f** Pearson correlation coefficients between normalized averaged patterns from 200 and 500 ms trials for all recorded neurons (black). Surrogate data are presented in gray (two-sided Mann–Whitney, *$p < 0.001$; statistical comparisons were calculated with observed and surrogated values from the 461 neurons that maintained stable conditions in 200 ms and 500 ms trials). **g** Average behaviorally evoked peri-event histograms for cells belonging to specific patterns for the 200 ms and 500 ms conditions. In each panel, the average lever trajectory (gray trace) is depicted as a reference. **h** Absolute number of cells ($n = 461$) expressing each pattern. **i** Percentage of cells belonging to each pattern. Percentages were calculated using sessions with more than eight simultaneously recorded cells. K-W values (df = 19, $X^2 = 41.29$, $p = 0.002$). The central line and box in boxplots (**f**, **h**) represent the median and 25th–75th percentiles, and whiskers extend to the most extreme datapoints excluding outliers.

as possible the two ISIs used in the passive stimulation experiments and to analyze two movements with at least twice the difference between them. Learning curves were constructed for the following movement parameters: (1) Intralimb correlation (Supplementary Fig. 6a) and its variance (Supplementary Fig. 6b), which indicate how stable the movement trajectory is over the course of learning. (2) Movement overshoot, which indicates movement duration (Supplementary Fig. 6c). (3) Movement effort (Supplementary Fig. 6d) and (4) movement speed (Supplementary Fig. 6e). Finally, we estimated three general performance values: the total number of trials performed in each session (Supplementary Fig. 6f), the inter-reward interval (Supplementary Fig. 6g), and the total amount of time to reach the first 50 trials, which is when animals are more engaged in the task (Supplementary Fig. 6h). All three values significantly improved with training. Altogether, our results demonstrate a robust behavioral protocol where neural data can be associated with two different movements matching our stimulation protocol in the temporal domain. Once animals were well-trained in the task, a craniotomy to access the VL/VM was performed (see methods). The following day, the animals resumed training for 3–4 days and were recorded (same as in the previous sections; one recording per session/day) while performing the behavioral protocol.

Under these conditions we were able to record 461 neurons from 13 mice during 36 sessions (2 to 4 recording sessions per animal). A great variability of activity patterns was observed when neural activity was aligned to the moment of reward delivery (Fig. 4c). These patterns were visible in both the 200 and 500 ms movement trials. Then, we explored if the spiking activity of individual neurons could be linearly associated with different parameters of movement execution, such as speed, overshoot, or amplitude, but found no evidence of such relationship (Supplementary Fig. 7; see methods). The activity of the cells as a population was organized as a gradient of activation/inactivation with at least two visible ends. On one end, a group of neurons decreased their firing activity during movement, and on the other end, a group of neurons visibly increased their spiking activity during movement (Fig. 4d). Interestingly, this structure remained almost unchanged in the 200 and 500 ms movement trials (Fig. 4d; both matrices were sorted from highest to lowest average firing rates in the 200 ms condition). When averaging neuronal activity in both types of trials (Fig. 4e), we divided the population dynamics into three phases: a rising phase that reached its highest peak before movement onset in either type of trial (indicated by color-coded arrows in Fig. 4d); a descending phase that was interrupted by the reward delivery; and a transitory, sharp response that was almost identical in both types of trials. To further investigate potential differences between neural activity in the 200 and 500 ms trials, for each neuron, we calculated the Pearson correlation coefficient between the average response during both trials. In this case, we normalized the temporal domain to have a similar number of bins in both conditions. Hence, we observed that most of the cells displayed high correlation values, indicating that the

spiking pattern was maintained independently of the movement duration (Fig. 4f). The distribution of correlation values was also compared with a surrogated distribution obtained by randomly circularizing the spiking activity of each neuron (see methods). Because average calculations over the whole population may mask the diversity of activation patterns (Fig. 4c), we used the same PCA/Silhouette-based method to detect specific optogenetic-evoked patterns in our previous sections (Fig. 2), but in this case to identify potential behaviorally related patterns. As before, we normalized the temporal domain to compare activity in both types of trials. Our analysis indicated that, based on Silhouette values (Supplementary Fig. 8a), neurons could be grouped into 10 types of patterns (Fig. 4g). Pattern type 1 was not particularly associated with an increase or decrease in activity; hence, we classified these neurons as non-related to the task. The remaining groups confirmed the presence of widely diverse activity patterns surrounding movement onset and reward delivery. With subtle changes, these patterns reflected subregions of the general architecture of the average population response depicted in Fig. 4e. Some patterns presented ramping activity with the highest peaks coinciding with movement onset (Fig. 4g; types 4, 6 and 7); one group presented ramping activity (Fig. 4g; type 2) or different levels of transient inactivation during movement maintenance (Fig. 4g; types 3, 8 and 9). Other neuronal groups exhibited increased activity coinciding with reward onset (Fig. 4g; types 5 and 10). Interestingly, while the highest Silhouette values were found for the 10-cluster projection, a four-cluster projection also presented significantly high values (Supplementary Fig. 8a–c). Both projections also matched the general architecture of the average population response. In terms of proportions, non-responsive neurons comprised the largest group, and the rest of the patterns were integrated by a similar number of neurons (Fig. 4h). To estimate if a particular pattern was preferentially associated with the 200 or 500 ms movement trials, we selected sessions where at least eight neurons were recorded simultaneously (24 out of the 36 sessions) and estimated the prevalence of each pattern in each session. We found that all patterns had a similar prevalence during the 200 and 500 ms trials (Fig. 4i). The fact that the two types of trials produced similar VL/VM population dynamics suggests that the duration for this type of movement is not particularly encoded in this region. On the other hand, the data also suggest that while different activation patterns could be formally extracted, it appears that thalamic neurons adjusted to three main phases: an ascending phase that ends around movement onset, a descending phase, and an abrupt increase triggered by reward.

## M1-evoked thalamic nPAPs are linked to specific movement-related activity patterns

Next, we investigated whether the neuronal activity that had been linked to different phases of movement could also be related to the preconfigured activation patterns triggered by M1 stimulation. To this end, immediately after each behavioral session, we administered

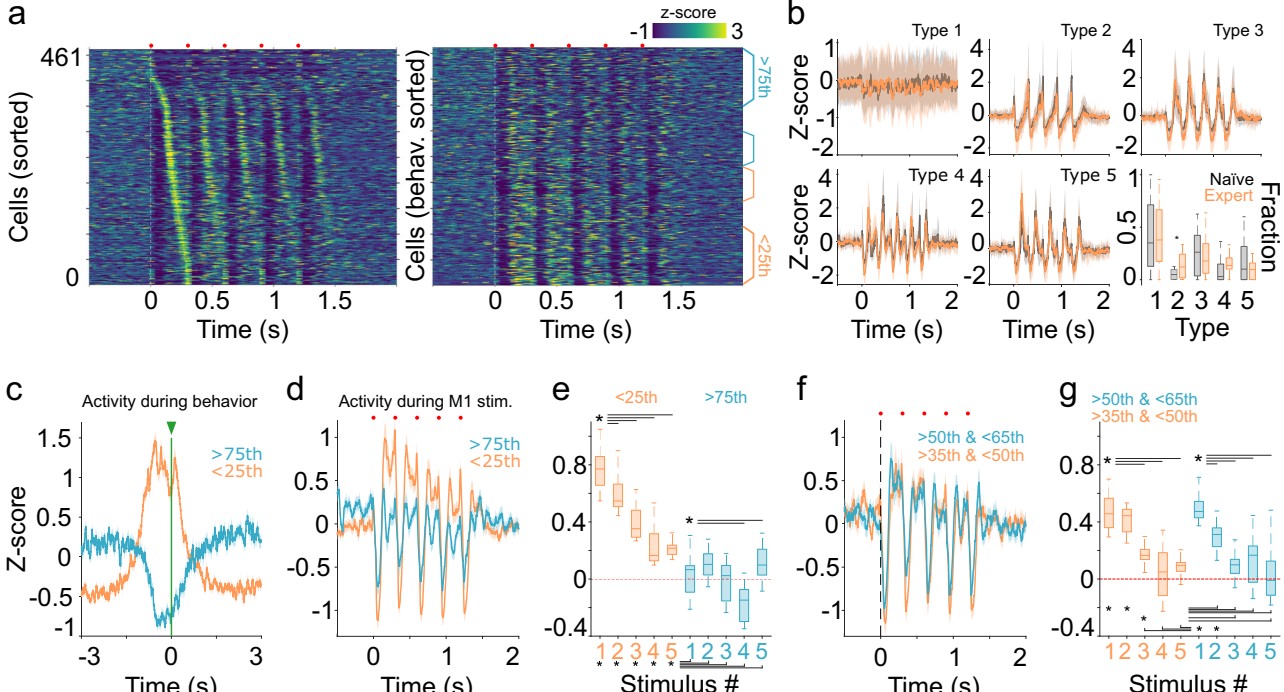

**Fig. 5 | Stereotypical thalamic patterns constrain movement-related activity.**
**a** Average firing rates (z-scored) evoked by M1 stimulation. Cells are time sorted according to their highest firing rate between the first and second stimulus of the train (left), or to the order obtained from behavioral execution displayed in Fig. 4d (right). **b** Average M1-evoked patterns for cells belonging to specific pattern clusters (300 ms ISI) in trained (orange) and naïve animals (gray). Pattern prevalence (bottom right panel, K-W, df = 9, $X^2 = 58.45$, $p < 0.001$; Type 2 $p = 0.019$). **c** Average firing rate during behavioral execution of two subgroups of cells above 75th and below 25th percentiles indicated in orange and blue brackets of panel a, respectively (aligned to reward delivery, green line and arrow). **d** Average firing rate evoked by M1 train stimulation (red dots) for the same subpopulations. Activity is aligned to the onset of the first stimulus of the train. **e** Amplitude of the subpopulation's late responses for each stimulus of the train

(K-W, df = 9, $X^2 = 1167.69$, $p < 0.001$; Bonferroni, <25th 1vs3-5 $p < 0.001$; >75th 1vs4 $p < 0.001$, 1vs5 $p = 0.002$; <25th 2–5 vs 75th 2–5 $p < 0.001$). **f, g** Same as in (**d, e**) but for subpopulations within the 50th to 65th and 35th to 50th percentiles of the population (indicated in a, right) (K-W, df = 9, $X^2 = 968.11$, $p < 0.001$; Bonferroni, <25th 1vs3-5 $p < 0.001$; >75th 1vs2,3,4,5 $p < 0.001$; <25th 1 & 2 vs 75th 2–5 $p < 0.001$; <25th 3 vs 75th 3, 5 $p < 0.01$; <25th 4,5 vs 75th 1,2 $p < 0.001$). Red dotted line in (**e, g**) as visual reference. Statistical comparisons in (**b, e, g**), were calculated in 461 neurons depicted in (**a**). In (**b–d, f**), solid lines and shaded areas represent the median and the 25th and 75th percentiles, respectively. Non-parametric one-way ANOVA was applied (**b, e, g**), and the central line and box in boxplots represent the median and 25th–75th percentiles, whiskers extend to the most extreme datapoints excluding outliers.

M1 stimulations identical to those used on our naïve animals (Figs. 1 & 2). In this way, we were able to identify behaviorally related and M1 stimulation-related activity in the same neurons. Because our previous analyses showed no differences between the 300 and 500 ms ISIs (Figs. 2, 4), for the following analysis, we focused on the 300 ms ISI train. VL/VM neurons recorded from expert animals displayed robust activation patterns in response to M1 stimulation, just like in the naïve animals (Fig. 5a, left panel). Then, to compare these patterns with those originally reported for the naïve animals, we pooled all neurons recorded from both groups (naïve and expert) and re-applied our PCA/Silhouette-based method to identify groups of neurons with similar response patterns. Once more, we found the five patterns observed in Fig. 2. Both groups of animals presented neurons in the five groups, further confirming the strong preconfigured organization of this cortico-thalamic interaction (Fig. 5b). However, we also observed that pattern type 2 was significantly more represented in the expert group (Fig. 5b, bottom right panel). Once it was confirmed that the neurons in the expert animals also displayed archetypical responses to M1, we explored potential crossings between behavioral and M1-related representations. In our first approach, we sorted the cellular activity evoked by M1 stimulation according to the order obtained from their activity during the behavioral sessions for the 200 ms movement trials displayed in Fig. 4c. This representation showed that cells that increased their firing rate during behavioral sessions appeared to produce more robust responses to M1 stimulation than cells that

decreased their firing rate during behavioral sessions (Fig. 5a, right panel). To confirm this possibility, we selected neurons located at both ends of the distribution produced during behavioral sessions (Fig. 4c), that is, cells below and above the 25th and 75th percentiles of the distribution, respectively. The average subpopulation response of these groups confirmed their characteristic activation and inhibition profiles during movement execution (Fig. 5c). But most importantly, when plotting their evoked activity during M1 stimulation, responses were clearly different between groups (Fig. 5d). Neurons with increased responses during behavior displayed higher amplitudes for the excitatory and inhibitory components of the M1-evoked response. Next, we analyzed if these two groups would also differ in terms of short-term adaptation to the stimulation train. To this end, we focused on the increase in long-latency firing rates (between 100 and 290 ms after each stimulus). We found that the group associated with increased responses during behavior presented a strong adaptation as the stimuli progressed (Fig. 5c, e; orange code), while the group associated with inhibitions showed little evidence of adaptation (Fig. 5c, e; blue code). This could be related to the fact that the responses of this group are mainly composed of pauses in activity. To confirm that these differences were related to the lower and upper ends of the behavioral distribution, we performed the same analysis but in cells from percentiles 35 to 50 and 50 to 65, where behaviorally related activity was more homogenous (Fig. 5a, right panel). In this case, both groups of neurons presented similar activation and

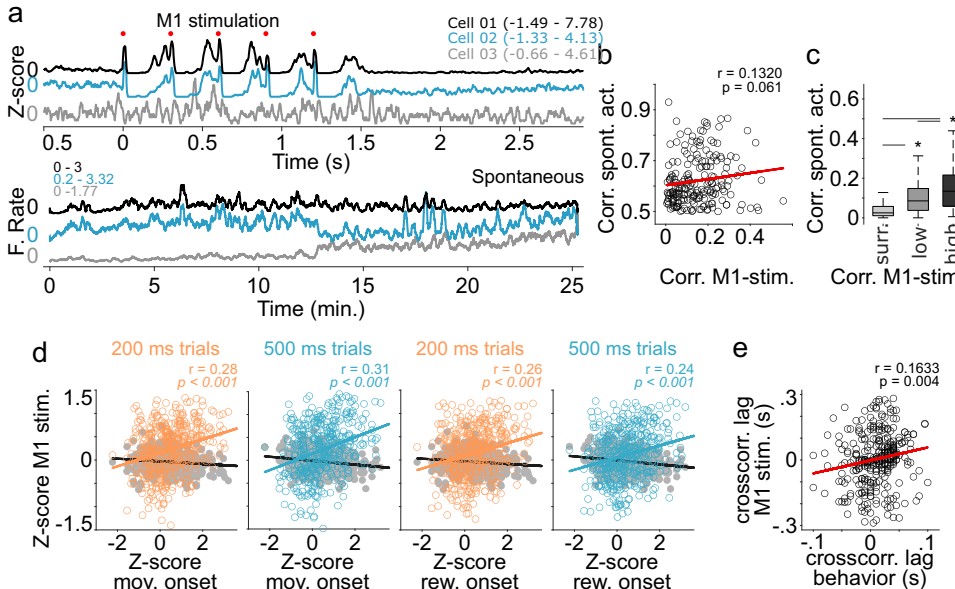

**Fig. 6 | Stereotypical evoked thalamic patterns and spontaneous activity.**
**a** Representative spike trains for three representative neurons (color coded) recorded simultaneously during the 300 ms ISI M1-stimulation protocol (Z-scored, upper panel) and during spontaneous activity (lower panel). The Z-score and firing rate ranges are presented for each neuron in each panel. Cells 1 and 2 were highly correlated in both, M1 stimulation and spontaneous conditions. **b** Scatter correlation plot for Pearson coefficients calculated during spontaneous activity and M1-stimulation protocols for all possible pairs of neurons recorded simultaneously. **c** Pearson correlation coefficients during movement execution between pairs of neurons with low or high correlation coefficients during M1 stimulation and for surrogate data (surr.) (K-W non-parametric one-way ANOVA; df = 2, $X^2 = 916.56$, $p < 0.001$; the central line and box in boxplots represent the median and 25th–75th

percentiles, and whiskers extend to the most extreme datapoints excluding outliers). Statistical differences are indicated by asterisks and lines joining specific comparisons (Bonferroni post hoc test, surr. vs low, low vs high and surr. high $p < 0.001$). **d** Scatter Pearson correlation plots between response amplitudes during movement execution around movement onsets (first and second panels) or reward onsets (third and fourth panels) and during M1 stimulation for the 200 ms (orange) and 500 ms (blue) movement trials. r and p values are indicated in each panel. Gray dots indicate correlations for surrogate data generated by randomly shuffling spike trains for the same neurons. **e** Scatter Pearson correlation plot for cross-correlation peak lags obtained from all possible pairs of neurons recorded simultaneously during movement execution and M1 stimulation. Pearson r and p values are indicated in (**c**–**e**).

adaptation patterns (Fig. 5f, g). Next, we explored the possibility of a more specific relationship between behaviorally related and M1-evoked responses. For this, we attempted to link PCA/Silhouette-based patterns obtained during behavior (the 10 pattern clusters from Fig. 4g) with those obtained by M1 stimulation (Fig. 5b; five pattern clusters). To perform statistical comparisons, we selected sessions with at least 10 simultaneously recorded neurons, and then we calculated the prevalence of M1 stimulation patterns for each of the behaviorally related patterns. While this high number of possible combinations decreased the statistical power, we were still able to detect that for behaviorally related pattern type 2 there was a significantly higher prevalence for M1-related pattern type 1 (Supplementary Fig. 8d). This result suggests that this ramping pattern type 2 is not related to M1-specific activation. On the other hand, behaviorally related pattern type 5 also displayed a significant bias for M1-related pattern type 4 (Supplementary Fig. 8d). These two patterns were characterized by similar shapes, with a decrease in activity followed by a rebound phase. This relationship further supports the notion that subgroups of thalamic neurons display constraining preconfigured dynamics.

On the other hand, previous reports in different brain regions have shown that evidence of preconfigured dynamics can be found in spontaneous activity[13,14,31]. To explore this possibility, we analyzed if pairs of neurons that were simultaneously recorded and presented high correlations during M1 stimulation (Fig. 6a upper panel, higher correlations than the 99.5 percentile of surrogated activity; see methods) would also present higher levels of correlation during spontaneous activity. Spontaneous activity was obtained in the same sessions than the behavioral and optical quantifications, but during periods where no stimulation or behavioral protocol was applied.

Under these conditions, we were able to extract 2145 pairs of neurons that were recorded in average 21.6 min (range 15.5 to 25.6 min) depending on the length of the session and the number of stimulation protocols and trials performed during the task. Pearson correlation coefficients were calculated from the spontaneous spiking activity of pairs of neurons discretized in 100 ms bins and smoothed with a Gaussian kernel filter of s.d. 200 ms (Fig. 6a lower panel). Then, while the correlation between correlation values during spontaneous activity and M1 stimulation was not significant (Fig. 6b), we found that the highly correlated pairs during M1 stimulation showed significantly higher correlation values during spontaneous activity than those from low-correlated pairs during M1 stimulation or correlation values obtained from surrogate spike trains (Fig. 6c). To further explore the existence of a latent organization, we wondered if the response sign (activation or inactivation), amplitude and latency to M1 stimulation could predict the same variables during movement execution. To this aim, for each neuron, we first calculated the z-score-based signed amplitudes to movement or reward onsets (positive and negative values corresponding to increases and pauses in activity, respectively). Then, we calculated the correlation coefficients between the response amplitudes obtained from the data aligned to movement onset (or reward onset) and the optogenetic stimulation onset and found significant correlations for both (Fig. 6d). After this, to determine a possible temporal relationship between neural activity during M1-stimulation and movement execution, for each pair of simultaneously recorded neurons we constructed cross-correlation histograms with the neural activity during movement onsets (from −2 to 2 s around movement onsets) and calculated the latency of the center of mass of each cross-correlogram. Then we calculated the difference in response latency between each pair of neurons during M1 stimulation.

At the end, the latencies from both conditions were correlated, rendering a significant relationship between the two variables (Fig. 6e).

Finally, it is possible that the different activity patterns may be also related to groups of neurons sharing common anatomical or molecular features, for example, different spike wave shapes potentially reflecting cell lineages, or neurons located in different thalamic regions (VL and VM). To address these possibilities, we performed a spike wave classification based on the first three principal components of the spike shapes plus the average firing rate of each neuron and their recording depth position. The best silhouette projection clustered spike waves into two very similar shapes. Firing rates (but not recording depth) were the main factor dividing the groups (Supplementary Fig. 9a–d). The first group comprised around 20% of the neurons (Supplementary Fig. 9e) and exhibited significantly higher firing rates than the second and more prevalent group (Supplementary Fig. 9f). Then, we calculated the probability that any spike shape would be associated with a particular response pattern to M1 stimulation, rendering two probability distributions, one for each spike wave shape. Finally, we created confidence intervals by randomly producing distributions from shuffled data neurons and sessions (1000 iterations). This analysis showed no statistical relationship between the two spike wave shapes and response patterns. Altogether, these observations further confirm the existence of thalamic nPAPs and that M1-evoked preconfigured responses are directly linked to thalamic responses associated with specific phases of forelimb movements.

The previously described phenomenon opens two questions. First, under which circumstances would these preconfigured responses be changed, modulated, or abolished? And second, would nPAPs modifications impact behavior? To start answering these questions, we studied the potential modulation of the SNr, a BG output nucleus and one of the main inputs to the VL/VM network over the stereotypical M1 VL/VM responses (Fig. 7a). Previous anatomical and functional observations indicate that SNr GABAergic projections modulate VL/VM activity with behavioral consequences[19]. Hence, we performed the same recording configuration (M1 VL/VM) but paired it with optogenetic manipulations of SNr GABAergic neurons (Fig. 7a). In principle, this manipulation could disrupt M1-evoked thalamic nPAPs and potentially associated behaviors. To explore this possibility, in a group of VGAT-Cre animals, we expressed ChR2 ($n = 4$) to stimulate SNr-projecting neurons. We found that SNr/M1-paired stimulation induced a disorganization of the VL/VM responses visible at the individual (Fig. 7b) and population levels (Fig. 7c). The stimulation diminished the sharp short-latency responses but, interestingly, maintained the inhibitory component of the M1-evoked responses (Fig. 7c, d). In particular, when comparing the amplitude of the short-latency component, SNr stimulation induced lower amplitudes in the three last stimuli of the train. This observation suggests that the stereotypical response is composed of a sharp M1 component, which is sensible to SNr modulation, and inhibitory and rebound effects, both insensible to SNr modulation. This was confirmed when we analyzed the amplitudes of the short- and long-latency components of the responses. In comparison to M1 stimulation alone, M1 stimulation paired with SNr stimulation induced a significant reduction of the sort-latency component of the response (Fig. 7e, left), especially the third-to-last stimuli of the train, but it produced no changes in the amplitudes for the long-latency responses (Fig. 7e, right). To explore potential differences in population dynamics, we constructed 30 bin matrices with the averaged activity of the five stimuli of the train, resulting in an averaged neural sequence for each experimental condition (Supplementary Fig. 10). Then, we compared the Euclidean distances between population matrices obtained from 300 vs 500 ms ISIs and the 300 ms condition with paired or unpaired SNr stimulation (Supplementary Fig. 10). The highest Euclidean distances were observed for the latter comparison, confirming that SNr modulation significantly modified the VL/VM population representation of the M1 stimulus (Fig. 7f). Next,

we explored if the average/population changes were related to a particular subgroup of neurons. To this end, we used our PCA/Silhouette-based classification method and selected the best projection with five groups. This projection yielded five almost identical response patterns to the ones obtained in the original analysis of Fig. 2 (Fig. 7g). In this case, SNr stimulation significantly decreased the occurrence of pattern type 2 and increased the occurrence of pattern type 4 (Fig. 7h). The former was characterized by neurons with sharp, short-latency responses, while the latter was characterized by strong pauses in activity. It is important to notice that, despite the differences, all patterns were present in the four animals recorded under these conditions (Fig. 7i). Importantly, M1-SNr co-activation did not produce different patterns but modified the prevalence of the existing ones. These results indicate that the preconfigured M1-VL/VM dynamics may be partially altered by the activation of the BG output nuclei. Hence, to evaluate the behavioral impact of disturbing VL/VM activity, we performed SNr stimulation (directly on the SNr, $n = 2$, or its terminals in VL/VM, $n = 2$; see methods) during movement execution in highly trained animals in our two-interval task (Fig. 7j; $n = 4$). We found that closed-loop activation triggered by minimum displacement of the lever induced significant decreases in movement stability (intralimb correlation) while sparing the rest of the variables (Fig. 7k; data was normalized to the median of control trials). These effects were similar for the 200 and 500 ms movements. Altogether, the previous section suggests that altering the preconfigured dynamics in VL/VM is sufficient to produce specific deficits in movement control, in this case, movement variability.

## Discussion

nPAPs have been described as organized activity parcelled into groups of neurons. But in the context of this work, involving inter-structure communication between M1 and VL/VM, an operational definition could be expressed as highly reproducible response patterns expressed by groups of individual neurons. Evidence of the existence of preconfigured neural dynamics has been previously reported for sensory cortices and cognitive regions, such as the hippocampus[2]. However, until now these dynamics have been only scarcely explored in motor networks. Here we focused on the motor thalamus of awake-behaving, head-fixed mice to address the possibility that preconfigured dynamics constrain movement and its related activity. Our results can be divided into two main sections: in the first, we demonstrate for the first time the existence of VL/VM thalamic nPAPs in response to M1 cortical stimulation. In the second, we described how those dynamics are linked to behaviorally associated thalamic activity. First, in naïve behaving mice that were briefly habituated to our recording conditions, we found that cortical inputs to the VL/VM evoked complex stereotypical responses organized as transient activations and inactivations with a variety of response latencies (Fig. 1). At first sight, these patterns were similar to the ones previously reported for the sensorimotor cortex, striatum, and SNr in anesthetized rats[9,11,32], suggesting a common responsive organization distributed throughout the thalamocortical-BG sensorimotor loops. These responses could be classified into four stereotypical patterns present in a large proportion of the VL/VM population and observed in virtually all awake (Fig. 2) and anesthetized (Supplementary Fig. 4) recorded animals. At the population level, these dynamics were organized as neuronal sequences spanning for a few of hundred milliseconds after cortical stimulation onset (Fig. 2) and resembled dynamics previously observed in cortical and subcortical sensorimotor systems[13,15,31,33], referred to as "information packets"[2]. However, contrary to what has been observed for sequential activations in the cortex and striatum of awake and anesthetized animals[6,16,34], and even in vitro preparations[34], VL/VM dynamics were unable to adapt to different temporal intervals (Fig. 2, Supplementary Fig. 1). This result suggests that VL/VM responses to M1 activation are rigid and further

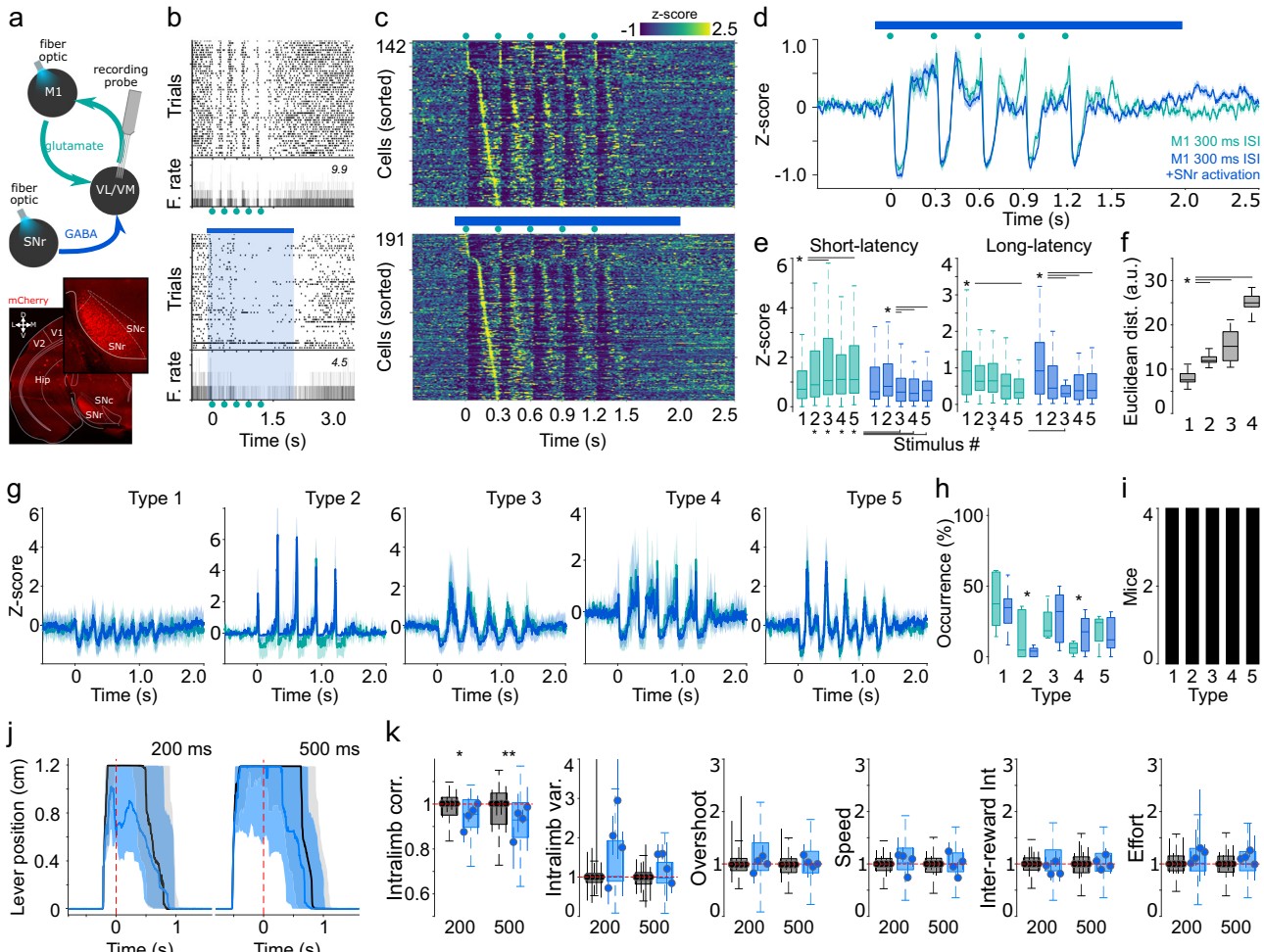

**Fig. 7 | Nigral modulation of thalamic preconfigured patterns and forelimb movements. a** Experimental configuration and histological confirmation of the stimulation sites in SNr. **b** Representative spike rasters and peri-event histograms for a VL/VM neuron with (bottom; blue throughout the figure) and without (top; green throughout the figure) associated SNr stimulation (blue bar/shade). M1 stimulation train (green dots) and maximum firing rate are displayed. Z-scored population matrices (**c**) and averaged population firing rate (**d**) evoked by M1 stimulation under the two conditions. **e** Response amplitudes for the short-latency (left; K-W, df = 9, $X^2 = 51.58$, $p < 0.001$; Short Latency, M1: 1vs3, 5 P = 0.038, 0.22; M1+SNr: 2vs3,4,5, $p = 0.019$, 0.009, 0.007) and long-latency (right; K-W, df = 9, $X^2 = 50.57$, $p < 0.001$; M1: 1vs4,5, $p = 0.001$, >0.001; M1+SNr: 1vs3,4,5, $p = 0.005$, <0.001, <0.001) components. **f** Euclidean distances between the following M1-evoked activation matrices: 300, 500 ms (absolute first 300 ms; **1**); 300, 500 ms (first 300 ms) with SNr-paired stimulation (**2**); 300, 500 ms (time normalized; **3**); 300 ms with and without SNr-paired stimulation (**4**). K-W values, df = 3, $X^2 = 100.95$,

$p < 0.001$; * 1 vs 2,3,4 $p < 0.0001$. Average shape (**g**), prevalence (**h**) and number of animals (**i**) displaying each M1-evoked pattern in both conditions (K-W, df = 9, $X^2 = 28.93$, $p = 0.0007$; type 2,4 $p = 0.04$, 4, $p = 0.03$). **j** Representative average lever trajectories aligned to reward delivery (red dashed line), for stimulated (blue) and non-stimulated (black) trials during 200 ms (left) and 500 ms (right) trials. Statistical comparisons in (**e**, **f**, **h**), were calculated in 146 neurons recorded in both stimulation protocols. **k** Behavioral variables in group (boxplots, $n = 4$) and individual animals (dots [median] and vertical lines [25th and 75th percentiles]; for intralimb correlation, K-W, df = 3; $X^2 = 13.26$; $p = 0.004$; *$p < 0.034$; **$p < 0.002$). K-W non-parametric one-way ANOVA was applied in (**e**, **f**, **h**, **k**), and the central line and box in boxplots represent the median and 25th–75th percentiles, and whiskers extend to the most extreme datapoints excluding outliers. Solid lines and shaded areas in (**d**, **g**, **j**), represent the median and the 25th and 75th percentiles, respectively.

supports their preconfigured nature. Then, we analyzed the anatomical origin of these responses and found evidence of CT neurons as main responsible for the general architecture of these dynamics, with a marginal component of PT neurons (Fig. 3). But what would be the consequence of this M1 stimulation-induced functional segregation? To answer this question, we showed that during the behavioral execution of a simple forelimb movement, thalamic activity can also be segregated into specific populations associated with distinct phases of movement (Fig. 4). But most importantly, specific subpopulations identified by their response pattern to M1 stimulation were associated with particular patterns of movement-related activity (Fig. 5). These results confirm that preconfigured functional segregation, revealed by passive stimulation of M1, constrains movement-related activity in the motor thalamus. However, it is not clear if this motor-related thalamic

activity is merely a reflection of M1 activity or, on the contrary, if it has specific effects on the final motor output. To clarify this point, we aimed to modulate preconfigured thalamic neural patterns by activating the SNr, a major inhibitory input to the VL/VM (Fig. 7). Interestingly, this manipulation was sufficient to modulate the early, short-latency, and excitatory components of the average population response and alter the occurrence of thalamic nPAPs. Furthermore, the same manipulation was also enough to significantly modify contralateral forelimb movement stability (Fig. 7).

We then aimed to determine the source of the segregated subpopulations described in this study. Although we did not anticipate a unique type of thalamic response to M1 stimulation, based on previous observations in cortical and basal ganglia regions[11], it is possible that the variability in response patterns would have been lower, as earlier

findings in the rodent motor thalamus indicate low heterogeneity in thalamocortical projection neurons[35] and little to no interneurons[36]. In this context, we found that four animals (Fig. 2h) expressed only one response pattern (that is, no other pattern was identified), opening the question on why some individuals may express less patterns than others. While we cannot rule out the possibility of an underlying structural difference in these animals, the fact that similar patterns were observed in the rest of the subjects, including naïve and trained animals (Fig. 5b), animals under urethane anesthesia (Supplementary Fig. 4), and in the isolated CT preparation (Fig. 3), suggest that this effect may be related to variability in our recording conditions, for example, variability in the final position of the optical fiber or recording probe. On the other hand, our results suggest that experience is not a factor in this segregation, since most animals in both the naïve and trained groups showed all the stereotypical patterns. Furthermore, these patterns were also observed under urethane anesthesia, demonstrating their brain-state independence. It is worth noting, though, that the anesthetized recordings were made after two awake sessions in which stimulation protocols were repeated hundreds of times. Even under these conditions, nPAPs remained stable. The possibility that these functionally segregated populations are experience-independent is consistent with recent studies performed in different non-motor thalamic nuclei. These studies have convincingly demonstrated the existence of genetically defined neural subpopulations associated with different anatomical profiles and functions[37,38]. For example, a recent study[39] showed that the reticular thalamic nucleus, typically conceived as the GABAergic homogeneous nucleus, is composed of at least two genetically defined subpopulations with distinct functional properties. Similar results have been observed in the parafascicular thalamic nucleus[40], where at least three independent thalamocortico-striatal motifs could be functionally and genetically identified. Previous literature demonstrates an unexpected level of neural heterogeneity that could at least partially explain our results in the VL/VM. In this context, while our analysis of the spike waveforms did not reveal special differences between response types, it is still possible that the different nPAPs evoked by M1 stimulation are associated with genetically distinct subpopulations, like those reported for the motor thalamus[41]. This possibility would further confirm the preconfigured nature of these signals and will be explored in future experiments. On the other hand, the thalamic nPAPs reported here may be triggered by two cortical subpopulations: pyramidal tract neurons known to project to multiple subcortical regions besides the motor thalamus, including the striatum and spinal cord and corticothalamic neurons projecting only to the thalamus[23]. In this work we started exploring the contribution of these two projections in naïve animals. Our data suggest that the general architecture of M1-VL/VM nPAPs is principally related to CT activation with a marginal component of PT (Fig. 3). However, future research will seek to address the exact contribution of these subpopulations not only to the nPAPs architecture, but its potential contribution to movement control. Furthermore, is also possible that other significant excitatory inputs to the VL/VM (e.g., cerebellar nuclei) influence these activity patterns by providing proprioceptive information that contributes to movement production in response to sensory inputs[27,42–44]. Another important finding was that we were able to correlate distinct M1-evoked patterns with specific activities associated with forelimb movements (Fig. 5, Supplementary Fig. 8). Previous studies in sensory cortical regions have demonstrated that spontaneously organized activity patterns (potentially preconfigured) are recruited by sensory stimulation[45]. More specifically, in a recent study, researchers observed that hippocampal neurons born on the same developmental day shared relevant encoding features during adulthood, such as anatomical connectivity patterns of even spatial representations[46]. Our data expands the interpretation of inside-out brain organization[47] from the cognitive to the motor sphere, at least for simple forelimb movements.

Our observations also raised another important question in the context of macro-circuit cortical, BG, and thalamic communication. In an attempt to answer this question, we came across several limitations in our approach. First, in most of our experiments, our optical stimulation strategy affected all cortical projections at the same time, including those projecting and not projecting to the motor thalamus. While this limitation was partially addressed by our cell-type specific manipulations for PT and CT neurons during M1-stimulation, their specific contribution during movement control is yet to be clarified. Second, we were limited by the lack of specific knowledge about the natural schemes of communication between the cortex and thalamus at the individual and population levels during the execution of a particular behavior. In future experiments, we will address this issue by recording cortical and thalamic activity simultaneously and adjusting cortical stimulation protocols to these natural dynamics. In this context, our selection of stimulation frequencies aimed to be a proxy of movement frequencies observed during forelimb movements in rodents when walking or trotting[9,25]. Moreover, the 300 and 500 ms ISIs used are consistent with goal-directed behaviors, such as reaching movements like those reported in this study[24,48], or even fast sequences of movements[49]. These frequency ranges have also been used in previous research under in vitro conditions[34] and in anesthetized animals[9,11] to study basic population dynamics in cortical and subcortical networks. On the other hand, although the input nucleus of the BG (i.e., the striatum) is massively innervated by the cortex, the BG exerts indirect influence over the cortex because of the thalamus (VL/VM). Hence, how do the different VL/VM neural patterns interact with BG signals to eventually impact behavior? Here we started exploring this interaction by directly activating SNr-to-VL/VM projections. To our surprise, the effect of this manipulation was almost restricted to the first part of the thalamic responses, the short-latency response (Fig. 7), resulting in a significant reduction in the occurrence of one neural pattern. This observation and previous reports studying sensory cortico-thalamic loops[23,50] suggest that the composition of M1-evoked thalamic nPAPs most likely involves a much more complex network of thalamic inputs, such as the reticular thalamic nucleus and/or the different cortical subpopulations projecting to the thalamus, both unexplored in this work. This manipulation was also associated with the disruption of movement variability, which could be considered a subtle behavioral effect; however, these results are consistent with two main lines of evidence from previous literature. First, M1 has been typically associated with motor plan production (i.e., establishing movement paths, directions, and trajectories[5,24,48,51]), while the motor thalamus has been proposed to gate and transmit information from subcortical circuits to maintain and re-organize cortical activity and associated motor commands[17,52]. Hence, it is unsurprising that VL/VM manipulation by SNr input activation affected the variability of movement but no other parameter, such as speed or direction. In fact, recent advances in BG literature suggest that one of the main roles of these nuclei, including output nuclei, is the modulation of commanded movements[24,32,49,53,54]. Our results, however, suggest that greater efforts will be needed to fully understand how the complex activity patterns observed at the individual and population levels in the striatum are translated into the VL/VM and, ultimately, to the motor cortex. For example, it is still difficult to picture how start, stop, speed, and time signals that are consistently recorded throughout the BG[6,9,25,55–58] are transformed or simply relayed by the VL/VM to the motor cortex. Our results indicate that upcoming signals from subcortical regions, such as the SNr, may interact with rigid, stereotypical patterns of thalamic activity. How these interactions are translated into organized motor commands or modulation of specific movement parameters will be the subject of future investigations. On the other hand, while we found that trained animals presented discrete differences with respect to naïve animals, the effects of learning on cortico-BG-thalamic dynamics must be thoroughly and explicitly explored in ad hoc experiments.

Furthermore, previous reports in sensory corticothalamic networks have shown different coding modes in response to repetitive whisker stimulation, suggesting that the current data set could be analyzed considering this framework[59]. It would also be interesting to understand how and when nPAPs are established during development. Finally, as suggested by previous literature[60], it would also be important to explore how pathological conditions, such as Parkinson's disease, may alter the functionality of these patterns and their interaction with cortical regions. Altogether, our results represent a proof-of-principle demonstration that inter-structure communication (in this case, M1–VL/VM) occurs based on hardwired preconfigured dynamics and perhaps also preconfigured anatomical connections, as reported elsewhere[61]. Deciphering these rules of communication throughout the cortico-BG-thalamic loops would be instrumental in understanding macro-circuit contributions to learning and execution of movements.

## Methods

All experiments were approved by the Animal Ethics Committee of the Institute of Neurobiology at the National Autonomous University of Mexico (UNAM) and conformed to the principles outlined in the Guide for the Care and Use of Laboratory Animals (National Institute of Health). Every precaution was taken to minimize suffering, and the number of animals used in the experiments.

### Animals

A total of 51 male mice were used in this study. From those, 40 animals were wild-type C57BL/6 and 11 were VGAT-Cre. Animals were 3- to 5-month-old male mice housed in individual acrylic boxes under temperature-controlled conditions with 12 h light/dark cycles and free access to food and water. Mice used for behavioral protocols (13 C57BL/6 and 4 VGAT-Cre) were water-restricted and consumed their requirements during training sessions (1 to 3 ml in 40 min per day). Animals were trained for six days with 24-hour free access to water per week. All experiments were conducted in the light phase of the cycle.

### Surgical procedures

All surgeries were conducted under aseptic conditions. Anesthesia was induced with a xylazine/ketamine cocktail (5/40 mg/kg) and maintained with sevoflurane (0.5–1.5%). A single subcutaneous dose of atropine was applied 10 min before surgery (0.025 mg/mg). The temperature and respiration were constantly monitored during surgery. After the surgical procedures, animals were supervised and administered antibiotics (gentamicine) and analgesics (meloxicam). *Stereotaxic viral infections*. All stereotaxic coordinates were calculated based on the Paxinos and Franklin mouse brain atlas and are reported in millimeters with respect to bregma. All injections were performed at a fixed rate (50 nl per minute) with a Hamilton micro syringe (NeuroSyringe 7001, 1 μL) and an infusion pump (3WPI-UMP3). The viruses AAV5/CamKIIa-hCHR2(H134R)-mCherry-WPRE-PA or AAV5/CamKIIa-hCHR2(H134R)-eYFP-WPRE-PA (purchased from VECTOR CORE, University of North Carolina) were injected unilaterally into two depths of M1 (AP: +1.34 mm ML: −1.75 mm DV: 1.25 and 1.75 mm) with a final volume of 1000 nl (500 nl in each deep). In VGAT-Cre mice, AAV5/EF1a-DIO-hCHR2(H134R)-mCherry or AAV5/EF1a-DIO-hCHR2(H134R)-eYFP (from VECTOR CORE, University of North Carolina), was injected into the SNr (AP: −3.52 mm ML: +1.5 mm DV: −4.0 mm) with a final volume of 400 nl. For experiments targeting PT neurons we injected the retrograde viruses AAV-pgk-Cre AAV into the VL/VM (AP: −1.0 mm ML: +1.1 mm DV: −3.25 mm) with a final volume of 400 nl and one month later we injected the pAAV-EF1a-doublefloxed-hChR2-eYFP into the DLS (AP: −0.14 mm ML: ±2.5 mm DV: −2.75 mm) with a final volume of 500 nl. pAAVretro-Syn-ChR2(H134R)-GFP; For experiments targeting CT neurons we injected the retrograde virus pAAVretro-Syn-ChR2(H134R)-GFP in the VL/VM (AP: −1.0 mm ML: +1.1 mm DV: −3.25 mm) with a final volume of 400 nl and the pAAV-flex-taCasp-

TEVp in M1 (AP: +1.34 mm ML: −1.75 mm DV: 1.25 and 1.75 mm) with a final volume of 1000 nl. AAV-pgk-Cre was a gift from Patrick Aebischer (Addgene viral prep #24593-AAVrg; http://n2t.net/addgene:24593; RRID:Addgene_24593). pAAV-EF1a-double floxed-hChR2(H134R)-EYFP-WPRE-HGHpA was a gift from Karl Deisseroth (Addgene viral prep # 20298-AAVrg; http://n2t.net/addgene:20298; RRID:Addgene_20298). pAAV-Syn-ChR2(H134R)-GFP was a gift from Edward Boyden (Addgene viral prep # 58880-AAV8; http://n2t.net/addgene:58880; RRID:Addgene_58880). pAAV-flex-taCasp3-TEVp was a gift from Nirao Shah & Jim Wells (Addgene viral prep # 45580-AAV5; http://n2t.net/addgene:45580; RRID:Addgene_45580).

*Fixation headpost implants*. Customized titanium headposts were implanted into the skulls of the animals, parallel to the midline. The headposts were fixed with dental cement, and two screws were placed on the left parietal bone that also served as ground and reference for the electrophysiological recordings (Fig. 1a). A metabond-based (C&B Parkell) clear-skull cap was built on the right hemisphere, and the craniotomy and fiber optic coordinates were marked. Craniotomies for silicon probe recordings were performed 24 h before the first recording session and protected with Kwik-Cast (WPI). Each animal was recorded in three to four sessions, one session per day. *Fiber implantation*. For optogenetic manipulation of the SNr, the same headpost implantation procedure described above was performed, but an additional 4 mm of optic fiber was directed to the SNr and fixed with dental cement.

### Optogenetic stimulation

Optic fibers (200 μm, diameter) for M1 and SNr stimulation were located in the following coordinates for M1: AP = 1.34 mm, ML = −1.75 mm, DV = between −0.3 mm and 1.0 mm; for SNr: AP = −3.52 mm, ML = 1.5 mm, DV: between −3.9 mm and −4.3 mm. For ChR2 excitation, we used blue light (465 nm) with a maximum power of 19.9 mW. In three animals, we tested different stimulation intensities ranging from 0.3 to 19.9 mW. For optogenetic stimulation of M1, we used four different train stimulation protocols. Trains consisted of five light pulses of 5 ms with 300 ms (3.3 Hz, protocol 1) or 500 ms (2 Hz, protocol 2). Protocols 3 and 4 were identical to protocols 1 and 2 but paired with a single continuous 2100 ms or 3400 ms (300 ms and 500 ms ISI, respectively) stimulus in the SNr. SNr stimulation started 100 ms before the first stimulus on M1 and covered the whole length of the train. *Stimulation during behavioral execution*. SNr optogenetic stimulation was delivered in 50% of the randomly selected trials. During stimulated trials, stimulation was configured as a closed loop triggered by the minimum displacement of the lever and terminated with the absence of movement or reward delivery.

### Behavioral protocol

*Apparatus*. Animals were trained and recorded in a head-fixed configuration in a customized stereotaxic-based behavioral set. Head-fixed animals rested on an acrylic platform with access to a holding pole and a movable lever located directly under their forepaws (Fig. 4a). The lever and holding pole were located 1.2 cm away from the platform. The lever was located under the left forelimb (contralateral to the recording and stimulation sites) and connected to a voltage transductor (1.2 cm = 2.5 mV). All voltage signals were digitalized and stored at 250 Hz through National Instruments cards. Water rewards were delivered through a water port connected to a solenoid valve. Rewards were signaled with a green LED located 10 cm in front of the animals. Two LEDs located 10 cm to the side of the head of the animal (one to the left, one to the right) were used to specify the duration of the required movement: blue light for long trials (500 ms) and white light for short trials (200 ms). All behavioral parameters were controlled and recorded with customized software programmed in LabView.

*Task*. Mice were trained in a two-interval behavioral protocol. Animals had to displace the left lever (push or pull) for at least 1.2 cm in

any direction and for at least 200 ms (short trials, indicated by a white LED) or 500 ms (long trials, indicated by a blue LED). Trials were self-initiated and presented in alternating blocks of 20 short or long trials. Once the spatiotemporal rule was achieved in each trial, the green LED indicated that the trial was correct, and the subject received a drop of water. After being rewarded, animals had to stop displacing the lever for at least 1.5 s before a new trial began. Sessions lasted 40 min and mice were trained in one session per day.

*Training.* Animals were accustomed to head fixation and general conditions for four habituation sessions where head fixation time was progressively increased from 5 to 40 min. After habituation, water restriction was started, and mice were progressively trained to displace the lever (from 50 ms to 500 ms) and associate the movement with reward (~10 sessions; modeling phase). After the modeling phase, formal training in the two-interval protocol started. All subjects were trained for at least 50 sessions before electrophysiological recordings or optogenetic manipulations were performed.

*Movement parameters.* The intralimb correlation was computed by calculating the Pearson correlation coefficient between each possible pair of trajectories for each category (200, 500 ms; Supplementary Fig. 6a), as well as its variance (Supplementary Fig. 6b). These parameters are indicators of how stable the movement trajectory is over the course of learning. For both parameters, we observed a quick improvement in the modeling sessions that remained stable during the rest of the training. We also observed that the 500 ms movements produced significantly lower correlation values and higher variances, especially in the early sessions of the two-duration phase, which is consistent with an increase in difficulty. Movement overshoot was defined as the amount of time that the animals maintained the movement after reward delivery. This measure indicates movement duration. We observed that animals presented similar overshoot values for both durations, rounding to about 200 ms (Supplementary Fig. 6c). Movement effort, an indication of movement efficiency, is the number of attempts that the animals make to obtain a single reward. To estimate this value, for each trial we quantified the cumulative amount of time that the lever was pressed to obtain the reward. This value decreased significantly with training, even for the more difficult 500 ms movement, which is consistent with motor improvement (Supplementary Fig. 6 d). Movement speed maintained similar values throughout the learning curve and different protocols (Supplementary Fig. 6e).

## Electrophysiological recordings
*General procedures.* Recordings were performed using silicon probe microelectrode arrays with a tetrode configuration (NeuroNexus; A4X4-tet-5mm-150-200-121). Craniotomies for recordings were located above the VL/VM complex (1.5 × 1.5 mm, centered at AP −1.34 mm, ML + 1.0 mm) and protected with Kwik-Cast (WPI). On recording day, Kwik-Cast was removed, and the array was impregnated with DiI for histological localization of the recording sites. Recording electrodes were inserted in the middle of the craniotomy with slight variations in the antero-posterior and mediolateral axes to avoid blood vessels. On different recording days, the probe was inserted in the same area. The electrodes were slowly lowered under sevoflurane anesthesia until reaching the desired depth (DV ~ 4.00 mm). Then, the animals were awakened and allowed to recover for at least 30 min before starting the recordings. Recordings on subsequent days were performed through the same craniotomy. After each session, the craniotomy was again protected with Kwik-Cast. To prevent infections and ensure the animals' wellbeing, recordings were limited to four sessions per animal.

*Electrophysiological data acquisition and processing.* Wide-band (0.1 to 8000 Hz) neurophysiological signals were amplified 1000 times via Intan RHD2000 series Amplifier System and continuously acquired at 20 kHz. Data visualization and processing were performed from raw data using Neuroscope and NDManager (http://neurosuite. sourceforge.net). Spike sorting was performed semiautomatically using the clustering software KlustaKwik (http://klustakwik. sourceforge.net) and the graphical spike-sorting application Klusters (http://klusters.sourceforge.net)[62,63].

## Analysis of neural data
Most of the recordings for M1-evoked thalamic activity patterns consisted of 50 stimulation trains, which were applied every 5 s (sometimes, 100 trains were applied; the average protocol duration was 4–8 min). Neurons were discarded when they presented no spiking activity or very low firing rates (<0.1 Hz) for 5 s (for M1-stimulation experiments) or 8 s (for behavior-related recordings) time windows around the onset of stimulation trains (or reward onset) in more than 40% of the stimulation trains. Due to recording stability conditions, for example, subtle silicon probe movements due to strong animal movements or the natural re-accommodation of the tissue, some neurons could only be recorded in one stimulation protocol or behavioral condition. These neurons were excluded from specific analysis when indicated in the main text. Firing rates were calculated based on inter-spike intervals over the entire recording period. Response latencies for the different components of the response (i.e., short- and long-latency increases and decreases) were calculated by binarizing neuronal data to 1 ms resolution and constructing peri-event histograms (−1 to +4 s around the onset of the first stimulus of the train). Then, for each cell, based on the 1 s baseline activity, we determined a confidence interval with an upper and lower limit of 99% (for increased responses) and 1% (for decreased responses), respectively. Responses to M1 stimulation were considered significant if they exceeded these limits by at least 1 ms. The duration of the first exceeding bin was defined as the response latency.

*Surrogated spike trains.* To construct random spike trains for the random distributions depicted in Figs. 2b, 4f, for each cell, each spike time from the spike trains obtained from the −1 to +4 s (or −4 to +4 s) around the onset of the stimulation (or reward delivery) time windows, was circularly and randomly re-ordered by adding time in a range of ±1 to 5 s (or ±1 to 8 s). Spike times offsetting outside the intervals were re-accommodated at the beginning or end of the interval depending on the magnitude and sign of the offset. This procedure was repeated 100 times. For surrogated spontaneous activity depicted in Fig. 6, we followed the same procedure but for the complete recorded spike trains and adding time in a range of a quarter of the length of the recording session.

*M1-evoked patterns.* We used the PCA/Silhouette-based method reported in refs. 11,32,64. In brief, we applied PCA to the z-scored M1 stimulation-evoked patterns of each VL/VM cell in a time window of 1.5 s or 2.5 s for the 300 ms or 500 ms ISI, respectively. To assign cells to specific clusters with similar characteristics, we applied k-means to the first three principal components. To obtain the best classification, we repeated the process 1000 times with projections ranging from 2 to 10 clusters. We scored each projection with the Silhouette method and selected the projections with the highest Silhouettes scores. This procedure detects the general shape of the patterns. To compare between pattern shapes (pattern clusters) evoked by the 300 ms and 500 ms ISIs (Fig. 2f), we focused only on the first 300 ms after each stimulus of the train for both conditions; that is, we cut the last 200 ms of the interval for the 500 ms condition. Then we pooled the activity of all recorded cells in both conditions and ran the procedure described above. The same classification method was applied for M1-evoked patterns from trained animals (Fig. 5b) and the experiments with paired SNr stimulation (Fig. 7g), except that in this case we compared the 300 ms condition from naïve vs expert animals, or the same animal but with or without SNr stimulation, respectively. Hence, it was not necessary to segment responses. We also used this classification method for the neural patterns associated with behavioral performance in the two-interval protocol (Fig. 4g; Supplementary Fig. 8) between the 200 ms and 500 ms movements. In this case, the temporal

domain was normalized to a fixed number of bins (2300). To this end, the 200 ms immediately before reward delivery for the 200 ms ISI condition was interpolated to 500 bins. This manipulation equalized the number of bins to 500 ms (500 bins) immediately before reward delivery in the 500 ms ISI condition. We complemented both conditions with 1000 ms (1000 bins) before the 200 ms (normalized to 500 bins) and 500 ms (500 bins) windows and 800 ms (800 bins) immediately after reward delivery, for a total of 2300 bins.

*Euclidean distance analysis.* To compare population dynamics between 300 ms and 500 ms ISI (Fig. 2) or between stimulation protocols with and without SNr-paired stimulation (Fig. 7), we calculated the Euclidean distances from population profiles obtained from the five stimuli's average M1-evoked responses. Euclidean distance analysis was based on the sorted 30-bin activity matrices in Supplementary Figs. 1, 10. In those matrices we calculated the Euclidean distance between each possible pair of bins under two conditions (300 ms ISI vs 500 ms ISI). We report the average Euclidean distance and the diagonal asymmetry index. For the diagonal asymmetry index, we first calculated two vectors: the minimum distance vector, with the distance values at the minimum, and the diagonal asymmetry vector, with the bin difference between the identity diagonal and the minimum values. The diagonal asymmetry vector was transformed into angular differences, where a difference of 30 bins between the diagonal and the minimum value corresponds to 360 degrees. For the self-sorted neural sequences where a condition was compared with itself (e.g., 300 ms ISI vs 300 ms ISI), the minimum distance and diagonal asymmetry vectors were constituted by thirty zeros (Supplementary Fig. 1d; first two matrices with dots falling exactly over the identity diagonal). In contrast, for cross-sorted sequence matrices, the minimum values of the distance matrix can vary between rows and columns, and both vectors were different from zero, as seen in the relative comparison between the 300 and 500 ms ISIs (Supplementary Fig. 1d).

*Generalized linear model analysis (GLM).* For this analysis, we used the five average pattern trajectories obtained from the PCA-based classification (reported in Fig. 2g) and an extra trajectory constructed with randomly generated data as neural data predictors. To this aim, we built 7 (columns) x n (rows) matrices in which the first six columns were occupied by each average pattern trajectory and the last column was occupied by the firing rate. Data were discretized into 10 ms bins, and trials were accumulated one after the other. Then, for each neuron we selected the best trajectory predictor (i.e., with the highest GLM regression coefficient) and generated five groups corresponding to the five different average pattern trajectories from the PCA-based classification (Supplementary Fig. 3a). Then, for each group of neurons we boxplotted the GLM regression coefficients for all predictors (patterns from PCA) and performed statistical comparisons (Supplementary Fig. 3b). If the highest coefficient truly corresponds to the identity of the classified group of neurons, it should statistically differ from the other predictors and from surrogated data generated from randomly shuffling the spike trains. Surrogated spike trains were generated as described above.

## Statistical analysis

Electrophysiological and behavioral data are presented as median +25th and 75th percentiles. Statistical comparisons for electrophysiological or behavioral data between groups were performed with Mann−Whitney or Kruskal−Wallis tests as stated in each section. A Bonferroni post hoc test was used for multiple comparisons. Statistical differences were considered significant if *P* values were <0.05.

## Histology

At the end of the experiments, mice were euthanized (pentobarbital 100–150 mg/kg) and transcardially perfused with 4% paraformaldehyde (PFA). The brains were collected and processed to confirm electrode, optic fiber, and infection sites.

## Reporting summary

Further information on research design is available in the Nature Portfolio Reporting Summary linked to this article.

## Data availability

The data supporting the main findings of this study are available at: https://doi.org/10.5281/zenodo.14042893. Raw datasets are available from the corresponding author upon reasonable request. Source data are provided with this paper.

## Code availability

Code used in this study is available at: https://doi.org/10.5281/zenodo.14042893.

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

## Acknowledgements

We thank Ana Inácio for indispensable technical advice and critical reading of this MS. Authors thank the support provided by all the members of Laboratory A-02 from the Institute of Neurobiology, UNAM; Cuautli Pacheco and Martín García for providing invaluable support in animal maintenance and care; Oscar Prospéro for generous donations of valuable equipment; Anaid Antaramian and Adriana González from Unidad de Proteogenómica, INB; and Jessica Gonzalez-Norris for proofreading. Perla González-Pereyra is a doctoral student from Programa de Doctorado en Ciencias Biomédicas, Universidad Nacional Autónoma de México (UNAM), and is supported by fellowship 749154 from CONAHCyT-México. This work was funded by grants: UNAM-DGAPA-PAPIIT: IN200822; IG200424 to PRO; CONAHCyT: FDC_1702, CF-2023-I-7 to PRO.

## Author contributions

Conceptualization: P.G.P. and P.E.R.O.; Methodology: P.G.P. and P.E.R.O.; Investigation: P.G.P., O.S.L., M.G.M.M., D.I.O.R., and P.E.R.O.; Data Curation: P.G.P. and P.E.R.O.; Formal analysis: P.G.P. and P.E.R.O.; Writing—original draft: P.G.P. and P.E.R.O.; Writing—review & editing: P.G.P., P.E.R.O., H.M., and L.T.; Resources: L.A.T. and H.M.; Supervision: P.R.O.; Project administration: P.R.O. and C.I.P.D.; Funding Acquisition: P.R.O.

## Competing interests

The authors declare no competing interests.
