## [Transparent Peer Review file · Nature Communications]

Preconfigured cortico-thalamic neural dynamics constrain movement-associated thalamic activity

Corresponding Author: Dr Pavel Rueda-Orozco

Version 0:

Reviewer comments:

Reviewer #1

(Remarks to the Author)

In this manuscript, authors describe that similar activity patterns can be evoked in the thalamus by optogenetic stimulation of M1 and by behavior. Thus, the authors claim that the thalamus has preconfigured activity patterns (nPAPs). This is interesting because it suggests that neurons in the thalamus have very constrained dynamics. Similar findings were reported previously in the sensory system, as discussed by authors. Thus showing sequential and constrained neuronal response also in the thalamus may help to better understand general principles of information processing in the brain. The main weakness of this manuscript is that authors only use one type of stimuli and one type of behavior to claim the existence of preconfigured activity patterns. However, this may be addressed with additional analyses (see below). Moreover manuscript in the current format has many redundant results that could be better moved to suppl materials, and additional analyses should be performed to give more insights into the nature of nPAPs.

Main points:

- 1) Redundant results. It is not surprising that neuronal responses are almost identical to very similar optogenetic stimuli. Thus, it would be better to only show results for 300 ms ISI, and combine Fig 2 and 3. The comparison to 500ms ISI could be moved to suppl. materials. Similarly, it is not surprising that neuronal patterns for 300ms and 500ms movement are also very similar. Again, it would be better to only show 300ms activity, combine fig 4 and 5, and move to suppl. materials all comparison to 500ms. The reason for this suggestion is that for the first 5 figures, I was just wondering what's new here because it is so obvious that similar stimuli or movements will evoke similar activity. The most interesting is Fig 6, and currently, the reader has to go through many obvious results first, before getting to Fig 6.
- 2) Analyses in Fig 6 are the most interesting and should be expanded. For example, it seems that neurons firing earlier to M1 stimulation also fire earlier during movement. This should be quantified. Authors can easily find the timing of the pick activity of each neuron and correlate those times between both conditions [like e.g. in ref 14]. Moreover, the authors mention that neurons with different types of activity may reflect different morphological/genetic types. Analyzing spike waveforms could help to address this question (for example fast-spiking interneurons have shorter spike waveforms than pyramidal cells [e.g. Barthó P, Hirase H, Monconduit L, Zugaro M, Harris KD, Buzsáki G. Characterization of neocortical principal cells and interneurons by network interactions and extracellular features. *Journal of neurophysiology*. 2004])
- 3) Comparison to spontaneous activity. The main weakness of this manuscript is that the authors only use one type of stimuli and one type of behavior. If nPAPs are really preconfigured in the thalamus then it should be also possible to detect such patterns in spontaneous activity. For example, authors can calculate cross-correlograms between neurons in the period preceding M1 stimulation and before behavior [see e.g. 14]. If e.g. neuron 1 tends to fire on average before neuron 2 during the spontaneous period, and similarly neuron 1 fires with shorter latency than neuron 2 to stimuli or in movement, then this would significantly strengthen authors claim about nPAPs.

Minor comments:

- 1) Fig 1f has no x and y axis
- 2) Fig 1gh; how latency was defined? What is a gray histogram?
- 3) Fig 1i – it is difficult to believe that long latencies are significantly different considering the overlap of both distributions. Maybe try a different non-parametric test to check if it is significant?
- 4) Information about statistical tests and p-values would be better inside of main text rather than in figure legends, as this would give more space to provide more details on stats.
- 5) Authors write: "We found that the group associated with increased responses during behavior." It is not clear to which neurons the authors are referring. Can you mark in fig 5 which are those neurons?
- 6) For SNr stimulation during behavior it would be good to show plots similar to Fig 7 b and c, to more directly compare both

activity patterns.

7) Why introducing a new term: nPAPs if this type of sequential activity is already known as neuronal packets? Even authors write that nPAPs: resembled dynamics previously observed in cortical and subcortical sensorimotor systems referred to as "information packets". Thus, introducing a new term only adds confusion if this phenomenon is already known under a different name.

8) Authors write about packets: "their existence and function in motor networks have not been explicitly studied". This is not exactly true, see for example: Xu W, de Carvalho F, Jackson A. Sequential neural activity in primary motor cortex during sleep. *Journal of Neuroscience*. 2019

Reviewer #2

(Remarks to the Author)

Through electrophysiological recordings during optogenetic control or behavioral tasks, the authors of this manuscript have elucidated the presence of a specific firing response called 'nPAPs' in the motor thalamus, which may contribute to motor behavior. This paper provides interesting results, but the reviewer has concerns about the functional meaning of this study.

Major

1. Conceptually, the motor cortex also involves sensory information processing (e.g., proprioception). Thus, the nPAPs in the motor cortex and thalamus may depend on other inputs, such as the DCN involving proprioception.

2. Optogenetic inhibition of cortico-thalamic inputs supports the conclusion that the nPAPs of thalamic neurons are associated with motor functions. There is information about inhibitory opsin in the Methods section, but there is nothing about inhibition experiments in the main text. If the authors had performed the inhibition experiments, the results would have been helpful in the interpretation of their results.

3. The authors mentioned that "300 ms and 500 ms ISIs produced indistinguishable population dynamics". However, in Figure 1i, why is there a difference in long-latency responses between 300 and 500 ms ISIs?

4. What is the significance of the differences between 'Sorted to 20 mW' and 'Self-sorted' in Supplementary Figure 2?

5. The authors mentioned that the pre-configured organization could be termed "information packets." Therefore, this reviewer expected to see in the manuscript the results of observing what specific "information" each movement-related thalamic pattern (or M1-evoked thalamic pattern) has and how these can contribute to specific components of animal behavior. However, the authors did not address such points. A more straightforward explanation will be that the stereotypical patterns appear just by the unique firing properties of neurons in the stimulated area.

- So, what roles does each movement-related pattern have? How does each pattern influence the behavioral variables involved in the task used in this study? Rather than simply observing the existence of patterns, it seems essential to explore and highlight which specific information they have.

- How are movement-related patterns and M1-evoked patterns related? It is not only M1 that could be associated with modulating the activity of thalamic neurons. The same would be true for modulating movements. Why did the authors take the M1 as the primary focus? Other brain regions, such as basal ganglia and reticular thalamic nucleus, might also play an essential role in thalamic nPAPs and the modulation of nPAPs according to behavior states. How about the results under the co-stimulation experiment on M1 and SNr inputs? that may cause different patterns in the VL/VM?

- Figure 6: M1-evoked pattern type 2 was high in the expert group. Supplementary Figure 5: M1-evoked pattern types 1 and 4 would be associated with behavior-evoked pattern types 2 and 5, respectively. Figure 7: M1-evoked pattern types 2 and 4 could be modulated by SNr stimulation. Considering all these, what role does each pattern type have?

6. It seems unsurprising that the M1-evoked thalamic responses are similar when the ISIs are different (unless the ISI is too short that stimulation is given once more before the response ends). Are there any reports of brain areas in which the response differs when given the trains of optogenetic stimuli with different ISIs? Also, in addition to ISIs and laser power, other temporal parameters, such as stimulation frequency or pulse width, could significantly impact the response. Have the authors conducted any tests on these matters?

7. In the Methods section, the authors needed to sufficiently explain how the various behavioral variables (lever position, intralimb corr., intralimb var., etc) were analyzed.

8. In Figure 7e, is it also necessary to verify the statistical significance between the M1 300 ms ISI condition and the M1 300 ms ISI + SNr activation condition to provide more robust support for the authors' findings?

9. The authors stated, "As a population, neurons presented baseline firing rates consistent with previous reports (median 3 Hz, 25th and 75th percentiles, 1.3 to 5.5 Hz)". For this reviewer, this seems somewhat low for the baseline firing of thalamic neurons. Are there any previous reports corroborating this?

Minor

1. In the representative image of Figure 1c, what is the reason for showing mCherry expression in M1 and eYFP expression in VL/VM? Are the two brain regions depicted in this representative image from different individuals?
2. What does the black line represent in Figure 2b (bottom)?
3. In Figure 4b, what is the aligning time point? Is it a reward onset?
4. Standardize the terminology to either ISI or IStI. In the main text, it is referred to as ISI, while in the Figure, it is labeled as IStI.
5. Some Supplementary Figures have titles, while others do not. Please display all titles.
6. The titles of Supplementary Figure 2 and Supplementary Figure 3 are the same. 2 appear to be data obtained from awake, and three appear to be from awake and anesthetized conditions. If the authors want to separate them into two figures, they have to consider the title so that it sufficiently represents each content.
7. In the peri-event histograms showing the firing rate, the y-axis labeling varies (for example, F.rate, Firing Rate, Rate, etc.). It would also be beneficial to add a scale bar or unit bar.

Reviewer #3

(Remarks to the Author)

In this manuscript, González-Pereyra et al. propose the existence of preconfigured activity patterns in the cortico-thalamic network. First, they use optogenetics to stimulate motor cortex (M1) in awake mice while recording in the motor thalamus (VL), and find that the impulse response in VL differs across neurons, and exhibits, on average, little adaptation to repeated stimuli delivered at 2 or 3.3 Hz. They classified the response types into five discrete groups, which had different patterns of excitation and suppression following the stimuli. Next, they recorded from VL neurons in animals performing a lever pull task, and observed a variety of firing patterns aligned to movement and reward. Cells with the largest behavior-related responses had larger responses to M1 stimulation, and these responses attenuated over repeated stimuli. Finally, stimulation of inhibitory projection neurons in the substantia nigra, pars reticulata suppressed the responses of some VL neurons to M1 stimulation, though this effect was small on average. Overall, the experiments were performed well, and the data are of high quality. Unfortunately, however, it is unclear what precisely the central claim of the manuscript is, how it is supported by the experimental evidence, and, more broadly, what the novelty and significance of the data are.

Conceptually, the principal weakness in this study is that the hypothesis, the experimental predictions it makes, and the observations that might falsify it are unclear. How exactly are preconfigured patterns defined, and what key features distinguish patterns that are preconfigured versus not preconfigured? If one stimulates brain area A, observes responses in brain area B, and finds that the responses in B neurons are correlated with their activity during behavior, is this sufficient to demonstrate that the behavioral patterns are preconfigured? Relatedly, I was unable to determine what the null hypothesis was, and what experimental observations would be predicted if the null were true. For example, would the authors predict that all VL neurons would have identical impulse responses if the dynamics were not preconfigured? Some heterogeneity in VL responses to M1 stimulation should occur based on the diversity of membrane time constants, synaptic inputs (including inhibitory inputs from the thalamic reticular nucleus), etc., and the range of responses the authors show is about what I would expect based on these considerations.

Many of the authors' conclusions rely on the classification of neurons into discrete groups based on their responses to M1 stimulation. This approach introduces several difficulties. (1) The classification into types appears to differ across experiments. For example, types 2-5 all look quite different in Fig. 3a in comparison with Fig. 6b. (2) The heatmaps seem to show a relatively continuous range of patterns, rather than a clustered structure. This raises the question of whether the types are genuinely distinct subsets of cells. (3) The responses of Type 1 appear to be highly variable across neurons, and it is difficult to distinguish between strong responses in single cells that cancel out after averaging, versus flat responses that are consistent across cells. Fig. 2a seems to show responses in most cells, but the non-responsive cells (Type 1) are reported to constitute 45-50% of the population.

To test for a relationship between the responses to M1 stimulation and activity during the behavioral task, the authors might consider modeling all neurons (e.g., with a GLM), in addition to the quantile-based slicing and averaging.

Photostimulation near a silicon probe can, under certain conditions, induce spike-like artifacts. The authors should address whether the responses with very short latency and low jitter, like those of the rightmost cell in Fig. 1d, might be contaminated by stimulation artifacts. It would be useful to see the raw data (voltage vs time), with spike times highlighted, within a short window of stimulation over several trials for this cell.

SNr stimulation has a robust inhibitory effect on the example neuron in Fig. 7b, but the effect on the population in the rest of the figure appears to be quite small.

There are a few typos in the figure legends (e.g., "Craneotomy" and "probe's trackt" in Fig. 1a, c).

Reviewer #4

(Remarks to the Author)
Review

The paper describes a research study on the responses of the motor thalamus to M1 stimulation in mice, using optogenetics, behavioral tasks, and various analytical methods, exploring the organization and functional significance of thalamic activity patterns. The paper focuses on the effects generated by two stimulation frequencies, and the potential role of these thalamic patterns in motor control and behavior.

By recording from the same neurons during both passive M1 stimulation and active movement execution, the researchers could directly compare the response patterns elicited in each context. They used principal component analysis (PCA) and silhouette-based clustering to categorize neurons into groups based on their response patterns, revealing correlations between the stereotypical patterns evoked by M1 stimulation and the diverse activity patterns related to movement execution. The study found that specific stereotypical response patterns elicited by M1 stimulation were directly linked to distinct phases of forelimb movements during the behavioral task. This linkage was evidenced by comparing the prevalence of response patterns during M1 stimulation with those observed during movement, showing that certain thalamic patterns are preferentially associated with specific movement-related activities.

This research is a tour-de-force, whose major message, in my view, is that the reductionist approach can work for thalamocortical functions in motor control. That is, studies like this, linking specific motor components to specific thalamocortical patterns, allow detailed systematic studies on behavior-relevant circuits with non-behaving and anesthetized animals. This empowers future research by expanding the scope of questions that can be asked, and tools that can be applied, in future studies.

My concerns are moderate, and do not reduce my enthusiasm from this impressive work.

Major

1. Previous studies on thalamocortical dynamics typically used more than two frequencies, showing that the frequency is an important factor. The authors thus should address this issue, explaining why they chose these specific frequencies, justify their choice in the context of motor control in this specific system and then discuss the limitations this choice induces.
2. The abstract is not clear enough in what are nPAPs, and specifically what is the nature of the principal nPAPs reported here. Please make an effort to better describe these aspects – it will facilitate the understanding of most of the readers.

Minor

1. In Fig. 1f, please describe what is shown exactly. A single cell? Population? What does the graph represent?
2. I suggest to address also the dynamics shown during the 5-stimuli trains. Previous studies (also involving thalamocortical circuits) showed that a stabilization process for response latency, in which longer latencies are associated with higher frequencies, such as shown in Fig. 1, might indicate an active closed-loop dynamics (phase-locked loop like). Although secondary to the aims of this research, such insights might be important for understanding thalamocortical dynamics during motor control.
3. Thalamic responses to passive stimulations show a kind of a slow wave superimposed on, or modulating, the faster stim-by-stim responses (e.g., Fig. 1j). This dual dynamics deserves a discussion and reference to previous relevant studies (e.g., Yu, Horev Cerebral Cortex 2015). I also suggest to test whether this slow dynamics is linked somehow (specific cells maybe) to the slow dynamics shown by thalamic neurons during movement execution (Fig. 5).

Version 1:

Reviewer comments:

Reviewer #1

(Remarks to the Author)

In the response letter authors say that they did requested analyses, however I do not see it in the manuscript. The main role of a reviewer is to help to improve manuscript, thus describing results only in the letter is not a proper revision.

As I wrote before in point #3: "The main weakness of this manuscript is that the authors only use one type of stimuli and one type of behavior. " I suggested to alleviate this problem by additional analyses to compare nPAPs to spontaneous activity. Authors agreed on value of those analyses by writing in the letter: "an approach like the one proposed by the reviewer would significantly strengthen our interpretations." However, despite describing results of those analyses in the letter, authors did not show it in the manuscript. In the main text authors mention "spontaneous activity" but it is unclear what they exactly did there. There is no information how "spontaneous activity" is defined, and plots 4 i&j, supposedly presenting analyses of spontaneous activity are showing something completely different: relation between activity evoked by M1 stimulation and behavior (at least this is what labels and legends describe).

Considering that current figure 4 is already crowded, maybe making a new figure showing sample activity during spontaneous period (e.g. few seconds before stimulation or before behavior) and presenting similar plots to fig 4h-j, where y-axis would be corresponding to spontaneous activity, could improve clarity of those analyses. The description of new analyses should also be significantly improved for clarity. As authors agreed that such analyses significantly strengthen interpretations, it would be also good to add a sentence also in the abstract that nPAPs have similar structure as

spontaneous patterns.
My other comments were largely addressed by authors.

Reviewer #2

(Remarks to the Author)
Remarks to the Author

The finding that M1 influences the thalamus aligns with previous hypotheses, suggesting its potential impact on lever press behavior is not entirely novel. Notably, the Karel Svoboda group has previously demonstrated a causal contribution of the thalamus in lever pressing. Introducing the concept of nPAPs to elucidate the role of cortico-thalamic circuits in movement might be novel, suggesting a new way to integrate information from the motor cortex and basal ganglia at the thalamus level. Nonetheless, the identification of nPAPs appears to be based solely on observing non-temporally adapting firing in the thalamus influenced by M1, raising the question of whether this is sufficient to assert the existence of nPAPs. As previously pointed out, I have concerns about the lack of enough functional data to support the conclusion of this work.

Reviewer's Comments

1) The authors' current optogenetic experiments lack specificity because they express and stimulate ChR2 in M1 while recording from VL/VM neurons. This approach limits the interpretation of their findings on the role of M1-VL/VM circuit in generating nPAPs. To address this limitation, the authors could inject a canine AAV-loxp-ChR2 virus into VL/VM thalamic nuclei and a CRE virus into M1. This strategy would enable selective M1-VL/VM circuit stimulation using optic fibers or optrodes implanted in the thalamus. Without this refinement, whether the observed nPAPs result solely from M1 input remains uncertain. In addition, the loss function optogenetics using halorhodopsin is necessary to support the input-specific roles in generating nPAPs.

-In the optogenetic experiment of SNR, they only have n=4 mice (SNR stimulation n=2 / SNR-VL/VM stimulation, n=2). They combined two groups to conclude. Additional experiments seem necessary to clarify the role of SNR.

2) The authors mentioned, "More importantly, we found that 12 out of the 18 animals exhibited the five patterns, two out of 18 presented four patterns, and only four animals presented only one pattern." If nPAPs function as information packets, each with behavioral meaning, how can it be explained that some individuals have more (5) or fewer (1) information packets?

- The number and portion of specific patterns must be described statistically. Otherwise, this numerical description is empirical and dampens any conclusion from this finding.

(a) Does an individual showing only one pattern have four other patterns for five?

(b) Or does it only have one?

What does each case (a and b) mean?

3) Figures 1 and 2 have virtually the same title. If the two figures reach the same conclusion, they should be combined into one figure. On the other hand, if the two figures reach different conclusions, their titles should be distinctly different.

4) As the authors mentioned in the main text, why do anesthetized mice still have stable preconfigured response patterns while there are no movements? Even in Supplementary Figure 4, there seems to be no difference in the occurrence and amplitude of response patterns between awake and anesthetized animals. How can it be concluded that nPAPs contain information about movement?

Minor Comments

In the main text, it is written, "Some patterns presented ramping activity with the highest peaks coinciding with movement onset (Fig. 3e; types 4, 6, and 7); one group presented ramping activity (Fig. 3g; type 2) or different levels of transient inactivation during movement maintenance (Fig. 3g; types 3, 8, and 9)". Shouldn't it be Fig. 3g and not Fig. 3e?

Many Supplementary Figures still lack titles.

It is recommended that two brain regions be represented, imaged from one individual, with eYFP expression, as shown in Figure 1c.

The peri-event histograms showing the firing rate have varying y-axis labeling (e.g., F. rate, Firing Rate, Rate). Adding a scale bar or unit bar showing firing rate values would also be beneficial.

Reviewer #3

(Remarks to the Author)

I appreciate the authors' revisions, which have improved the manuscript and addressed several of my concerns. The paper provides valuable information about the effects of M1 stimulation on thalamic activity, and I think this is a strength. However, the central claim of preconfigured activity patterns still relies on a vague definition of these patterns and indirect experimental evidence for their existence, which limit the rigor, conceptual advance, and expected impact of the study. The authors take

inspiration from compelling studies of precise spike timing in the hippocampus, where sequential activation of place cells is recapitulated, temporally compressed, and often reversed during sharp wave ripples. The unambiguous demonstration of replay (Foster & Wilson, Nature 2006; Diba & Buzsaki, Nat. Neuro. 2007) depended critically on (1) a precise operational definition of the phenomenon (the sequence of cells that spike during SWRs should be the same as the sequence during track traversals, or should be the same sequence in reverse), (2) visualization of clear examples of individual replay events, and (3) an analysis of the distribution of a statistic computed within individual events (in the cited studies, the rank correlation between spiking order). While subtle statistical challenges remain (see Tingley & Peyrache, Proc. Royal Soc. B, 2020), the existence of SWR replay has been clearly established, and has had a major impact in the hippocampal field and beyond.

In my opinion, the current study does not meet these three criteria. I am still unable to determine what kinds of patterns are and are not preconfigured, and whether the former can be identified in these data. The authors should state clearly what they mean by preconfigured patterns - is a preconfigured pattern a specific sequence of neural firing, which occurs both during natural behavior and in response to stimulation of an input? If so, analyses should be focused on testing for a concordance between firing sequences in the two situations. The closest analysis in the current manuscript is in Fig. 4j, which shows a correlation between the time of cross-correlation peaks for pairs of cells during stimulation and reaching. This correlation is relatively weak, and a null distribution does not seem to be computed (though it is for the amplitudes in 4h). More importantly, this is an indirect way of determining the similarity of temporally-extended stimulation- and reach-aligned firing sequences. If the authors show heatmaps of reach-aligned activity, sorted by the stimulation-aligned firing order, is a diagonal structure present? Additional analysis methods, such as those previously used to identify hippocampal replay, or those used to examine low-dimensional cortical dynamics in primate reaching studies, could enable the authors to really focus on the central issue of how similar the stimulation- and reach-related patterns are.

Reviewer #4

(Remarks to the Author)

The authors addressed all my concerns. Congratulations for an excellent scientific contribution.

Version 2:

Reviewer comments:

Reviewer #1

(Remarks to the Author)

Authors largely addressed my concerns.

Minor comments:

- 1) In sentence "previous reports in different brain regions have shown that evidence of preconfigured dynamics can be found in spontaneous activity." authors should provide references to those papers.
- 2) In the new figure some results are marked as statistically significant with "**", however in main text there is mention what were p values and what test was used. Please provide this information.

Reviewer #2

(Remarks to the Author)

Congratulations. This finding significantly contributes to our understanding of the thalamus's role in integrated sensory-motor information processing. I have no further comments.

Reviewer #3

(Remarks to the Author)

I appreciate the authors' revisions, which have further improved the manuscript. However, while it provides valuable data about the effects of motor cortical stimulation on thalamic spiking in the mouse, I remain unconvinced that these data exhibit a novel phenomenon of preconfigured patterns in the motor system. The definition of such patterns as "highly reproducible response patterns expressed by groups of individual neurons," which includes responses evoked by electrical or optogenetic stimulation, is extremely broad, and could apply to the results of almost any stimulation experiment. Numerous studies over the past sixty years have applied microstimulation to one sensory or motor area while recording single-unit discharges in another, and have shown the temporal properties of these discharges can vary across neurons. This is fully expected based on differences in intrinsic excitability, synaptic strength, conduction delays, and similar considerations.

To put it differently, I am still unable to determine what the null hypothesis was. If the experiments were designed to test for heterogeneity in thalamic responses to cortical stimulation (which the authors interpret as preconfigured patterns), what plausible experimental outcomes would have provided a negative answer?

Point-by-point response

Preconfigured cortico-thalamic neural dynamics constrain movement-associated thalamic activity

Perla González-Pereyra, Mario G. Martínez-Montalvo, Diana I. Ortega-Romero, Claudia I. Pérez-Díaz, Hugo Merchant, Luis A. Tellez, Pavel E. Rueda-Orozco*.

REVIEWER COMMENTS

Reviewer #1 (Remarks to the Author):

In this manuscript, authors describe that similar activity patterns can be evoked in the thalamus by optogenetic stimulation of M1 and by behavior. Thus, the authors claim that the thalamus has preconfigured activity patterns (nPAPs). This is interesting because it suggests that neurons in the thalamus have very constrained dynamics. Similar findings were reported previously in the sensory system, as discussed by authors. Thus showing sequential and constrained neuronal response also in the thalamus may help to better understand general principles of information processing in the brain. The main weakness of this manuscript is that authors only use one type of stimuli and one type of behavior to claim the existence of preconfigured activity patterns. However, this may be addressed with additional analyses (see below). Moreover, manuscript in the current format has many redundant results that could be better moved to suppl materials, and additional analyses should be performed to give more insights into the nature of nPAPs.

R:

We thank the reviewer for time spent revising our MS and the helpful comments; addressing them have significantly improved the clarity of our findings.

Main points:

1) Redundant results. It is not surprising that neuronal responses are almost identical to very similar optogenetic stimuli. Thus, it would be better to only show results for 300 ms ISI, and combine Fig 2 and 3. The comparison to 500ms ISI could be moved to suppl. materials. Similarly, it is not surprising that neuronal patterns for 300ms and 500ms movement are also very similar. Again, it would be better to only show 300ms activity, combine fig 4 and 5, and move to suppl. materials all comparison to 500ms. The reason for this suggestion is that for the first 5 figures, I was just wondering what's new here because it is so obvious that similar stimuli or movements will evoke similar activity. The most interesting is Fig 6, and currently, the reader has to go through many obvious results first, before getting to Fig 6.

R:

Our reasoning was to describe as thoroughly as possible the thalamic population dynamics associated with M1 stimulation. However, we acknowledge the validity of the reviewer's observation, and as a result, we have significantly restructured the presentation of these figures in accordance with the recommendations.

On the other hand, we recognize that the 300 and 500 ms interval stimulation protocols were similar. However, we decided to explore both protocols because prior research on neural dynamics in motor areas has shown that a 200 to 300 ms interval is sufficient to significantly modify the population trajectory architecture associated with rhythmic movements (e.g. Reference 29 in our original MS, Betancourt et al, 2023). It has also been demonstrated that similar time intervals (100, 300, 500 ms) have produced significantly different dynamics in cortical networks cultivated *in vitro* (Goel & Buonomano et al, 2016; Reference #36 in our original MS). Nonetheless, we agree that displaying only the 300 ms interval in the main figures enhances the clarity of the results. As suggested, we have moved the comparisons with the 500 ms analysis to supplementary materials and combined Figures 2 and 3.

2) Analyses in Fig 6 are the most interesting and should be expanded. For example, it seems that neurons firing earlier to M1 stimulation also fire earlier during movement. This should be quantified. Authors can easily find the timing of the pick activity of each neuron and correlate those times between both conditions [like e.g. in ref 14]. Moreover, the authors mention that neurons with different types of activity may reflect different morphological/genetic types. Analyzing spike waveforms could help to address this question (for example fast-spiking interneurons have shorter spike waveforms than pyramidal cells [e.g. Barthó P, Hirase H, Monconduit L, Zugaro M, Harris KD, Buzsaki G. Characterization of neocortical principal cells and interneurons by network interactions and extracellular features. Journal of neurophysiology. 2004]

R:

As suggested by the reviewer, we have included the following analyses: First, we aligned the neural activity to movement onsets (or reward onsets) and calculated the activity peak latency of each neuron with respect to those onsets. Because many neurons presented different levels of activation or inactivation triggered by M1 stimulation or movement or reward onsets, we also calculated the amplitudes of such responses. Then, we calculated the correlation coefficients between the peak latency or response amplitudes obtained from the data aligned to movement onset (or reward onset) and optogenetic stimulation onset. We found significant correlations for both movement, and reward onsets, but the former were slightly higher. These correlations were compared with correlations obtained by applying the same analysis in surrogate spike trains. Together with our previous analyses, these observations further confirm the relationship between cortico-thalamic nPAPs and movement dynamics. These data are now included in the new Figure 4H and in the main text.

Regarding the second part of the reviewer's remark, previous reports describe the rodent motor thalamus as a homogeneous region (Bosch-Bouju; Kaneko; Jager), without

interneurons. For this reason, we did not originally explore this possibility. However, following the reviewer's advice, we performed two types of spike wave classifications, one using only the first three principal components of the spike shapes and another including the same three principal components plus the average firing rate of each neuron and their recording depth position. We found that the data was better fitted when including the firing rate. The best silhouette projection clustered spike waves in two very similar shapes which firing rates (but not the recording depth) were the main factor dividing the groups. The first group was integrated by around 20% of the neurons and presented significantly higher firing rates than the second and more prevalent type. Then, we calculated the probabilities of any spike shape to be associated to a particular response pattern to M1 stimulation, rendering two probability distributions, one for each spike wave shape. Finally, we created confidence intervals by randomly producing distributions from shuffled data neurons and sessions (1,000 iterations). This new analysis showed no statistical relationship between any of the two spike wave shapes and response patterns. We interpreted these results as a partial confirmation of our previous speculation, that is, nPAPs are more related to the pattern of connectivity inputs that each group cluster receives than intrinsic thalamic populations. However, we cannot discard the possibility that neurons with different genetic profiles may express very similar spike wave shapes. We have now included these two analyses in a new supplementary figure (Supplementary Fig. 8) and in the results and discussion sections.

3) Comparison to spontaneous activity. The main weakness of this manuscript is that the authors only use one type of stimuli and one type of behavior. If nPAPs are really preconfigured in the thalamus then it should be also possible to detect such patterns in spontaneous activity. For example, authors can calculate cross-correlograms between neurons in the period preceding M1 stimulation and before behavior [see e.g. 14]. If e.g. neuron 1 tends to fire on average before neuron 2 during the spontaneous period, and similarly neuron 1 fires with shorter latency than neuron 2 to stimuli or in movement, then this would significantly strengthen authors claim about nPAPs.

R:

The main objective of our study was to explore the existence of preconfigured dynamics in motor networks and the motor thalamus. In our first approach, we focused on controlling well-known principal inputs to this region. However, an approach like the one proposed by the reviewer would significantly strengthen our interpretations.

We addressed this observation in two ways. First, we calculated Pearson correlation coefficients between average peri-event histograms of every possible pair of neurons that were recorded simultaneously. After this, we calculated the Pearson correlation coefficient between the same pairs of neurons during spontaneous activity periods (when no stimulation was provided). Then, we created confidence intervals for spontaneous and M1-evoked correlations based on the shuffled spike times of the same neurons (1000 iterations). This approach revealed that neurons that presented higher correlation values during M1 stimulation also presented higher correlation values during spontaneous activity. These correlation values were significantly higher than those obtained from pairs

with low correlation values during M1 stimulation and the 99.5 percentile of the surrogated distribution.

Then, following the reviewer's advice, we calculated the cross-correlograms between all possible pairs of simultaneously recorded neurons and established the latency of the center of mass of each cross-correlogram. On the other hand, we calculated the difference in latency between each pair of neurons during M1 stimulation. Finally, latencies from both conditions were correlated. We found a slight but significant relationship between them, suggesting the existence of a fixed structure in the neural activation.

Minor comments:

1) Fig 1f has no x and y axis

We have now included scale bars in the figure.

2) Fig 1gh; how latency was defined? What is a gray histogram?

This is stated in the methods section and are defined as follows: Response latencies for the different components of the response (i.e., short- and long-latency increases and decreases) were calculated by binarizing neuronal data to 1 ms resolution and constructing perievent histograms (-1 to +4 s around the onset of the first stimulus of the train). Then, for each cell, based on the 1 s prior baseline activity, we determined a confidence interval with an upper and lower limit of 99% (for increased responses) and 1% (for decreased responses), respectively. Responses to M1 stimulation were considered significant if they exceeded these limits by at least 1 ms, and the time of the first exceeding bin was defined as the response latency. As stated in the figure legend and right Y axis in both panels, gray histograms in Fig. 1g, h correspond to latency to the inhibitory component of the response, called "decrease".

3) Fig 1i – it is difficult to believe that long latencies are significantly different considering the overlap of both distributions. Maybe try a different non-parametric test to check if it is significant?

In Figure 1i, we applied a Mann-Whitney test for individual 300 vs 500 ms comparisons rendering a significant difference with $p = 0.0305$. However, as suggested by the reviewer, we applied a different non-parametric test (Kruskall-Wallis) including short-latency, decrease, and long-latency comparisons. This comparison produced no significant differences (post hoc, bonferroni or least significant difference). We have modified the figure and text accordingly.

4) Information about statistical tests and p-values would be better inside of main text rather than in figure legends, as this would give more space to provide more details on stats.

We have included the statistical information in the figure legends based on comments from previous rounds or revisions.

5) Authors write: "We found that the group associated with increased responses during behavior." It is not clear to which neurons the authors are referring. Can you mark in fig 5 which are those neurons?

We have now clarified this point in the main text.

6) For SNr stimulation during behavior it would be good to show plots similar to Fig 7 b and c, to more directly compare both activity patterns.

R:

Unfortunately, we were not able to perform simultaneous recordings with stimulation during behavioral execution. A much more complete work on the sub-cortical inputs and their influence in behavioral control and M1-VL/VM interactions will be the subject of future investigations.

7) Why introducing a new term: nPAPs if this type of sequential activity is already known as neuronal packets? Even authors write that nPAPs: resembled dynamics previously observed in cortical and subcortical sensorimotor systems referred to as "information packets". Thus, introducing a new term only adds confusion if this phenomenon is already known under a different name.

R:

We agree with the reviewer. In our opinion, the activity we observe here clearly reflects the concept of "information packets." Our intention was not to introduce a new concept but to support the one proposed before. Since information packets have been mostly linked to sensory and cognitive processing, our intention was to use a more descriptive term.

8) Authors write about packets: "their existence and function in motor networks have not been explicitly studied". This is not exactly true, see for example: Xu W, de Carvalho F, Jackson A. Sequential neural activity in primary motor cortex during sleep. Journal of Neuroscience. 2019

R:

We have modified this and included the example reference.

Reviewer #2 (Remarks to the Author):

Through electrophysiological recordings during optogenetic control or behavioral tasks, the authors of this manuscript have elucidated the presence of a specific firing response called 'nPAPs' in the motor thalamus, which may contribute to

motor behavior. This paper provides interesting results, but the reviewer has concerns about the functional meaning of this study.

Major

We thank the reviewer's attention and observations. In the following part we will address these observations.

1. Conceptually, the motor cortex also involves sensory information processing (e.g., proprioception). Thus, the nPAPs in the motor cortex and thalamus may depend on other inputs, such as the DCN involving proprioception.

R:

We completely agree with the reviewer. We believe these nPAPs are the result of the complex anatomical patterns of connectivity. In our study, we focused on the motor thalamus, which integrates both excitatory and inhibitory inputs. Excitatory inputs are mainly provided by at least two classes of neurons from the primary motor cortex M1 (pyramidal tract and cortico-thalamic neurons) and the cerebellum. , bringing proprioceptive information that contributes to movement production in response to sensory inputs (Vitek et al., 1994; Middleton & Strick, 2000; Boch-Bouju et al., 2013). Inhibitory inputs arrive from the output nuclei of the basal ganglia, substantia nigra pars reticulata, the internal segment of the globus pallidus, and the reticular thalamic nucleus. In future research, we will try to disentangle the specific contribution of all potential inputs to the motor thalamus. In this work, we started exploring these possibilities as seen in our original Figure 7, now Figure 5. There, we activated inhibitory inputs from the SNr, and found that this manipulation only modified the sharp and initial component of the population thalamic response, supporting the notion that NPAPs are the result of the activation of complex anatomical connectivity patterns. We have included these arguments in the discussion.

2. Optogenetic inhibition of cortico-thalamic inputs supports the conclusion that the nPAPs of thalamic neurons are associated with motor functions. There is information about inhibitory opsin in the Methods section, but there is nothing about inhibition experiments in the main text. If the authors had performed the inhibition experiments, the results would have been helpful in the interpretation of their results.

R:

We apologize for this mistake. We are not reporting experiments with inhibitory opsins. This is now corrected in the methods section.

3. The authors mentioned that “300 ms and 500 ms ISIs produced indistinguishable population dynamics”. However, in Figure 1i, why is there a difference in long-latency responses between 300 and 500 ms ISIs?

R:

Yes, we found statistical differences in the latency of the late component of the thalamic response and differences in the amplitude of the short and long components. However,

as can be seen throughout the MS, the general architecture of the response is reliable. As mentioned in the results, these differences may be related to variability in the short-term adaptation of individual neurons. To support this possibility, in our early recordings, we applied 100 ms trains in addition to the 300 and 500 ms stimulation trains to four animals (10 recording sessions) included in this study. In the following figure (**Fig. R1**), we display the activity of three representative cells with typical response patterns of short-latency activation, followed by a brief pause and a final rebound. As can be seen, in the 100 ms protocol, the rebound is absent in the first four stimuli because the cycle is interrupted by the following stimulus of the train. However, in the last stimulus of the train (indicated by red arrows), the rebound is clearly higher than in the 300 ms or 500 ms trains, confirming a different level of adaptation depending on the stimulation protocol. It is also important to notice that, in the case of the 300 and 500 ms trains, the typical architecture is maintained but the amplitudes of the components differ.

To address this fair concern, in the main text we have replaced the word “indistinguishable” with “similar dynamics”. On the other hand, Reviewer #1 also pointed out this statistical difference in the amplitude of long-latency responses (Minor comments point #3). The reviewer commented on the possibility that this difference was not statistically significant and proposed we use a different non-parametric statistical method. In our original approach, we applied a Mann-Whitney test for individual 300 vs 500 comparisons rendering a significant difference with $p = 0.0305$ for long-latency amplitudes. However, as suggested by the reviewer, now we applied a Kruskal-Wallis test including short-latency, decrease, and long-latency comparisons, and found no significant differences with different post hoc methods. We have modified the figure and text accordingly. It is important to highlight that this point does not change the particular or general conclusions of our work.

Figure R1. Representative spike rates and their corresponding average peri-event histograms for three neurons (one neuron in each row) recorded in VL/VM. Activity was aligned to the first stimulus of the train (indicated by red dots at the top). The five stimuli of the stimulation train were given at 10 Hz (left column), 3.3 Hz (middle column) and 2 Hz (right column) and are indicated above each plot by red dots. Red arrows highlight the “rebound” long-latency response to the last stimulus of the train as a visual contrast between stimulation frequencies.

4. What is the significance of the differences between 'Sorted to 20 mW' and 'Self-sorted' in Supplementary Figure 2

Neuronal activity can be sorted according to the moment of their peak firing rate in each stimulation protocol, which is what we call “self-sorted”. Alternatively, the matrices produced by each power can be sorted according to times of the peak firing rates in a particular condition, in this case 20 mW stimulation. We decided to plot both ways

because it could happen that the different powers of stimulation produce a different neuronal sequential order. Alternatively, the sequences could be the same in every condition independently of the power used. As can be appreciated in the figure, sorting all matrices by the order obtained in the 20 mW condition produced similar activation sequences, indicating that nPAPs start appearing after about 2.5 mW and remain stable through high powers of stimulation.

5. The authors mentioned that the pre-configured organization could be termed “information packets.” Therefore, this reviewer expected to see in the manuscript the results of observing what specific “information” each movement related thalamic pattern (or M1-evoked thalamic pattern) has and how these can contribute to specific components of animal behavior. However, the authors did not address such points. A more straightforward explanation will be that the stereotypical patterns appear just by the unique firing properties of neurons in the stimulated area.

We thank the reviewer for the opportunity to clarify this point. In our introduction and discussion, we used the term “information packets” as a direct reference from previous literature where discrete sensory stimulation (somatosensory, olfactory or auditory) induced similar sequential activations to the ones reported here. In those cases, the type of information is assumed to be sensory, and the specific modality is dependent on the region and structure.

In this work, we decided to record the motor thalamus; hence, the information would be related to motor commands arising from the M1. As depicted in our Figure 5b-c and described in the main text, the average thalamic neural activity around reaching movements could be divided into three main phases: a rising phase that reached its peak just before movement onset; a descending phase that was interrupted by the reward delivery giving place to a short peak in activity. Then, a transitory, sharp response that was almost identical in both types of trials. When exploring the activity of individual neurons, we observed that neurons were grouped into clusters reflecting sub-components of the average response (Figure 5,e). Neurons belonging to clusters type 2, 4, 6 and 7 presented ramping activity, while clusters, 3, 8 and 9 produced pauses in their activity during movement maintenance. Finally, clusters 5 and 10 presented peaks in activity triggered by the reward onset.

Thus, our original objective was coherent with the reviewer’s concern: what kind of motor information would neurons classified as part of M1-evoked clusters (Fig. 6b) carry? The first and most evident relationship is that neurons that increased their firing rates during behavior presented clearer M1-evoked patterns than those neurons that decreased their spiking activity during behavior (Fig. 6C-d). This reviewer’s comment is in a way similar to the first reviewer’s comments. We have now included a new series of analyses showing that neurons with higher correlations during M1 stimulation are also more correlated during behavioral execution (Please see the modifications in our original Fig. #6, now #4).

- So, what roles does each movement-related pattern have? How does each pattern influence the behavioral variables involved in the task used in this study? Rather than simply observing the existence of patterns, it seems essential to explore and highlight which specific information they have.

Neurons belonging to clusters type 2, 4, 6 and 7 presented ramping activity before movement onsets. Neurons belonging to clusters 2, 3, 8 and 9 produced ramps (2) or pauses (3, 8 and 9) in their activity during movement maintenance. Finally, clusters 5 and 10 presented peaks in activity apparently triggered by the reward onsets. Unfortunately, this is the only type of relationship we can extract. Ideally, manipulating each cluster during movement execution would help to causally link them to each movement phase. For example, artificially increasing the activity of clusters 3, 8 and 9 during movement maintenance would probably shorten this phase. Unfortunately, we do not have any tool to manipulate specific clusters. To achieve this goal, we would first need to disentangle what are the structural components related to each M1-evoked pattern, something that is outside the limits of this study, but in the scope of our following research objectives.

To highlight this relationship between neural activity patterns and behavior, we have modified the main text and our original Figure 5g (now Fig. 3g) including the average movement trajectory together with average neural pattern.

- How are movement-related patterns and M1-evoked patterns related? It is not only M1 that could be associated with modulating the activity of thalamic neurons. The same would be true for modulating movements. Why did the authors take the M1 as the primary focus? Other brain regions, such as basal ganglia and reticular thalamic nucleus, might also play an essential role in thalamic nPAPs and the modulation of nPAPs according to behavior states. How about the results under the co-stimulation experiment on M1 and SNr inputs? that may cause different patterns in the VL/VM?

We agree with the reviewer. Aside from the different subpopulations of M1 neurons projecting to the motor thalamus, these regions also receive excitatory inputs from the cerebellum and inhibitory inputs from at least the SNr and reticular thalamic nucleus. We started exploring the existence of nPAPs in the M1-thalamic connection because M1 is the main connection to the thalamus, but as the reviewer mentioned, the architecture of nPAPs is most likely related to the different inputs of VL/VM. This possibility is supported by the results in the M1/SNr co-stimulation experiment. There we found that the general architecture of the VL/VM response was disrupted, particularly in the amplitude of the short-latency component of the response. Consistently with this, SNr activation significantly decreased the prevalence of pattern type 2 (original Fig. 7G, now Fig. 5g), characterized by prominent short-latency increases in spiking activity. This manipulation also increased the prevalence of pattern type 4, characterized by pauses in activity. Both effects would be consistent with the activation of an inhibitory input. Finally, it is important to notice that SNr co-activation did not produce different patterns but modified the prevalence of the existing ones. We have complemented the main text with these arguments.

- Figure 6: M1-evoked pattern type 2 was high in the expert group. Supplementary Figure 5: M1-evoked pattern types 1 and 4 would be associated with behavior-evoked pattern types 2 and 5, respectively. Figure 7: M1-evoked pattern types 2 and 4 could be modulated by SNr stimulation. Considering all these, what role does each pattern type have

In our original text we stated that our analysis to match behavioral-related and M1-related patterns was limited by the possibility to simultaneously record enough neurons. Because of those limitations, we cannot establish the specific role of each M1-evoked pattern during movement. With that said and despite these limitations, we were able to observe that behavioral pattern #5 was significantly linked to M1-related pattern #4. Behaviorally related pattern #5 consisted of a decrease in spiking activity during movement maintenance followed by a sharp increase in activity at reward delivery. Neurons that expressed this pattern exclusively expressed the M1-related pattern #4, consisting of a very similar shape, a brief decrease in firing rate followed by a sharp rebound. These results indicate that at least in a subpopulation of VL/VM thalamic neurons, M1-evoked preconfigured responses are associated with a specific part of the behavioral response, in this case, the maintenance and reward phases.

On the other hand, neurons that expressed the behaviorally related pattern #2 (ramping activity interrupted by reward onset) presented a higher prevalence and almost exclusive prevalence for M1-related pattern #1. This pattern was considered as a non-significant response to M1; hence, this type of ramping activity appears not to be related to a specific response to M1.

We have now highlighted these arguments in the discussion section and modified Supplementary Fig.00 to make these relationships more explicit.

6. It seems unsurprising that the M1-evoked thalamic responses are similar when the ISIs are different (unless the ISI is too short that stimulation is given once more before the response ends). Are there any reports of brain areas in which the response differs when given the trains of optogenetic stimuli with different ISIs? Also, in addition to ISIs and laser power, other temporal parameters, such as stimulation frequency or pulse width, could significantly impact the response. Have the authors conducted any tests on these matters?

R:

This is a similar comment to the first comment of Reviewer #1. We decided to make these comparisons because previous reports in S1-cultivated neurons adapted the latency of the responses according to the duration of an interval (two electrical pulses separated by 100, 250 or 500 ms), suggesting an intrinsic dynamic capable of temporal adaptation (Goel & Buonomano, 2016).

On the other hand, as mentioned in response #3, in early experiments we tested the 100 ms ISI frequency. Because the 100 ms ISI interrupts the full response of neurons presenting rebounds after 100 ms, this protocol was not continued for the following experiments. We did not originally test for different pulse widths, but the test for different powers (original Supplementary Fig. 3, now 00) showed similar responses.

7. In the Methods section, the authors needed to sufficiently explain how the various behavioral variables (lever position, intralimb corr., intralimb var., etc) were analyzed.

R:

Based on this reviewer's comment and that of reviewer #1, we have now expanded our explanation of the different behavioral variables in the methods section.

8. In Figure 7e, is it also necessary to verify the statistical significance between the M1 300 ms ISI condition and the M1 300 ms ISI + SNr activation condition to provide more robust support for the authors' findings?

R:

We thank the reviewer for this suggestion. We performed the suggested comparison and SNr stimulation induced significantly lower amplitudes in the last three stimuli of the train. We agree with the reviewer that this comparison provides a more robust support to our claims. We have modified the figure and main text accordingly.

9. The authors stated, "As a population, neurons presented baseline firing rates consistent with previous reports (median 3 Hz, 25th and 75th percentiles, 1.3 to 5.5 Hz)". For this reviewer, this seems somewhat low for the baseline firing of thalamic neurons. Are there any previous reports corroborating this?

R:

We are now including the appropriate reference. In (Bosch-Bouju, et al 2014; Jneurosci) the authors report very similar firing rates to the ones reported in our work.

Minor

1. In the representative image of Figure 1c, what is the reason for showing mCherry expression in M1 and eYFP expression in VL/VM? Are the two brain regions depicted in this representative image from different individuals?

R:

We apologize for the lack of clarity. We used constructs with either mCherry or eYFP. In Figure 1, for example, we chose eYFP because it creates a better contrast with the dye to stain the micro electrode arrays, emitting in the same channel as mCherry. We have now clarified this point.

2. What does the black line represent in Figure 2b (bottom)?

R:

The black traces represent surrogate response patterns obtained from spike time stamp randomization. We have now clarified this point in the new Figure 2.

3. In Figure 4b, what is the aligning time point? Is it a reward onset?

R:

The reviewer is correct. Movement trajectories are aligned to reward onsets indicated by vertical green lines. We have now clarified this point in the figure legend.

4. Standardize the terminology to either ISI or IStl. In the main text, it is referred to as ISI, while in the Figure, it is labeled as IStl.

R:

We have now standardized these abbreviations.

5. Some Supplementary Figures have titles, while others do not. Please display all titles.

R:

We have now included titles in all supplementary figures.

6. The titles of Supplementary Figure 2 and Supplementary Figure 3 are the same. 2 appear to be data obtained from awake, and three appear to be from awake and anesthetized conditions. If the authors want to separate them into two figures, they have to consider the title so that it sufficiently represents each content.

R:

We apologize for this mistake. As the reviewer mentioned, Supplementary Figure 2 corresponds to awake recordings while Supplementary Figure 3 corresponds to anesthetized recordings. We have modified the figure legends accordingly.

7. In the peri-event histograms showing the firing rate, the y-axis labeling varies (for example, F.rate, Firing Rate, Rate, etc.). It would also be beneficial to add a scale bar or unit bar.

R:

We have now included unit references.

Reviewer #3 (Remarks to the Author):

In this manuscript, González-Pereyra et al. propose the existence of preconfigured activity patterns in the cortico thalamic network. First, they use optogenetics to stimulate motor cortex (M1) in awake mice while recording in the motor thalamus (VL), and find that the impulse response in VL differs across neurons, and exhibits, on average, little adaptation to repeated stimuli delivered at 2 or 3.3 Hz. They classified the response types into five discrete groups, which had different patterns of excitation and suppression following the stimuli. Next, they recorded from VL neurons in animals performing a lever pull task, and observed a variety of firing patterns aligned to movement and reward. Cells with the largest behavior-related responses had larger responses to M1 stimulation, and these responses attenuated over repeated stimuli. Finally, stimulation of inhibitory projection neurons in the substantia nigra, pars reticulata suppressed the responses of some VL neurons to M1 stimulation, though this effect was small on average. Overall, the

experiments were performed well, and the data are of high quality. Unfortunately, however, it is unclear what precisely the central claim of the manuscript is, how it is supported by the experimental evidence, and, more broadly, what the novelty and significance of the data are.

Conceptually, the principal weakness in this study is that the hypothesis, the experimental predictions it makes, and the observations that might falsify it are unclear. How exactly are preconfigured patterns defined, and what key features distinguish patterns that are preconfigured versus not preconfigured? If one stimulates brain area A, observes responses in brain area B, and finds that the responses in B neurons are correlated with their activity during behavior, is this sufficient to demonstrate that the behavioral patterns are preconfigured? Relatedly, I was unable to determine what the null hypothesis was, and what experimental observations would be predicted if the null were true. For example, would the authors predict that all VL neurons would have identical impulse responses if the dynamics were not preconfigured? Some heterogeneity in VL responses to M1 stimulation should occur based on the diversity of membrane time constants, synaptic inputs (including inhibitory inputs from the thalamic reticular nucleus), etc., and the range of responses the authors show is about what I would expect based on these considerations.

R:

We thank the reviewer for the time spent in revising our work.

We based our definition of preconfigured patterns on previous literature, mainly from sensory processing and hippocampal regions. As stated in the introduction, our main objective was to explore the existence of such patterns in motor networks, and in case of finding evidence supporting this possibility, to explore their potential implication in behavioral control. The significance of such findings would support and expand to motor brain regions, current hypotheses from the sensory and cognitive domains (observed in hippocampus and sensory cortices) indicating that preconfigured neuronal activity would be the scaffolding of more complex representations in this case, movement-related signals.

We recognize that this definition and the evidence supporting it is fundamental. Among the most relevant features typically attributed to preconfigured networks are a non-random structural organization that can be reflected in organized patterns of neural activity, such as neural sequences. These types of activation can be observed under different behavioral states, such as awake, sleeping or anesthetized conditions. In this study, we found similar dynamics in response to M1 stimulation in naive and expert awake animals and also in anesthetized conditions. This concern is similar to the second comment from reviewer #1. In that comment, the reviewer suggested that if the activity was preconfigured, then it would be possible that the activity of cells that are correlated during M1 stimulation would also be correlated during spontaneous activity. To address this possibility, we performed a new series of analyses (now reflected in Fig. 4H-j) that confirm higher levels of correlation during spontaneous activity between pairs of neurons that present higher correlation values during M1 stimulation (To avoid repetition, we politely ask to revise our response to reviewer #1).

On the other hand, we did not predict or assume that all VL neurons would have identical impulse responses. As mentioned in the introduction, based on previous reports from our group performed in different cortical and basal ganglia regions, we were expecting a certain level of variability in the response patterns. For example, sensory cortex and SNr presented fewer patterns than the dorsolateral striatum in response to optic stimulation of sensory pathways (Peña-Rangel et al, 2021, ref #31 in the original main text). However, it is also true that the rodent motor thalamus has been reported as a homogeneous structure, with little variability in the type of excitatory neurons and little to no interneurons. This could have resulted in fewer response patterns to the five evoked by M1 stimulation or the 10 observed during movement execution. We have now included these arguments in the discussion.

Many of the authors' conclusions rely on the classification of neurons into discrete groups based on their responses to M1 stimulation. This approach introduces several difficulties. (1) The classification into types appears to differ across experiments. For example, types 2-5 all look quite different in Fig. 3a in comparison with Fig. 6b. (2) The heatmaps seem to show a relatively continuous range of patterns, rather than a clustered structure. This raises the question of whether the types are genuinely distinct subsets of cells. (3) The responses of Type 1 appear to be highly variable across neurons, and it is difficult to distinguish between strong responses in single cells that cancel out after averaging, versus flat responses that are consistent across cells. Fig. 2a seems to show responses in most cells, but the non-responsive cells (Type 1) are reported to constitute 45-50% of the population.

R:

We thank the reviewer for this observation that helped us to clarify the presentation of the data.

(1) As we shown in Supplementary Figure 5, our method is statistically robust. However, based on the reviewer's remarks, we have complemented this method with a GLM (please see below). Regarding point (2) in this comment, we agree with the reviewer that a basic visual inspection of the heat maps would suggest a continuous range of patterns. This effect is expected for any matrix sorted according to moment of the highest peak of their firing rate in a fixed time window in this case, the 300 ms interval between stimuli 1 and 2 of the train. In this kind of representations, any neuron displaying spiking activity within the interval will have a moment with a maximum firing rate, giving the impression of a continuous sequential activation. However, this perception of continuity contrasted with two basic observations. First, the same heat maps also show that in upper and lower ends, the patterns are clearly different. Second, when inspecting the diversity of raster plots from individual neurons, it was clear that many would not present clear responses to the optic trains and many others would present diverse response patterns (Original Fig. 3B, now Fig. 2g). However, as mentioned before, neurons with no clear responses would

necessarily have a moment with a highest rate, and this would also be reflected in the perception of the sequential activation. We reasoned that this kind of representation may be masking the existence of clusters of neurons with similar dynamics. Hence, we opted for the PCA-based method of classification. To help clarify the reviewer’s concern, in the next figure (**Fig. R2**) we are plotting the original 300 ms sorted matrix vs. the matrix without the type 1 group of cells and a matrix only composed of type 1 neurons. As can be seen, the latter matrix (bottom panel) presents only a “thin” sequence in the first stimulus of the train, confirming the artificiality of the sequential perception. However, the matrix composed of neurons classified as types 2-4 demonstrated robust and repeatable sequences in all stimuli of the train. It is also important to notice that it appears more defined than the original matrix encompassing all neurons (types 1-5).

Figure R2. a) Average firing rates (z-scored) evoked by M1 stimulation for cells recorded under the 300 ms inter-stimulus interval (ISI). Cells were sorted according to the moment of their highest firing rate between the first and second stimulus of the train. Each stimulus of the train is indicated on top of each panel (green dots). Upper, middle and lower panels display, all recorded neurons, neurons classified as Type 2 to 4 and neurons classified as Type 1, respectively.

Concerning point (3), it is true that one possibility is that pattern type 1 may be the result of highly variable patterns canceling each other. But, according to our data, this is not the case. The previous plot started to clarify this point. However, to further address this issue, in the following figure (**Fig. R3**) we show the average response of every single cell belonging to each cluster depicted our original Figure 3a (now Fig. 2f). This representation demonstrates that, as mentioned in our main text, type 1 classified units presented no evident responses, without any evidence of variable classifiable patterns averaging each other. On the other

hand, the same representation for neurons classified as types 2 to 5 provides a visual reference of the robustness of our PCA-based classification method.

Figure R3. Average responses to the 300 ms ISI for every neuron reported in Fig. 2a and Supplementary Fig. 01. Neurons were divided into five types as classified in Fig. 2F and displayed in five corresponding panels. On each panel, each trace corresponds to the average response of one neuron. Traces were aligned to the first stimulus of the train and each stimulus of the train is indicated above each panel by green dots.

To test for a relationship between the responses to M1 stimulation and activity during the behavioral task, the authors might consider modeling all neurons (e.g., with a GLM), in addition to the quantile-based slicing and averaging.

R:

We have performed the suggested analysis. For this, we have applied GLM and used the average pattern trajectories obtained in our PCA-based method (from original Fig. 3B, now 2g) as predictors for neural activity. By extracting the maximum GLM coefficients, we found that neurons that expressed maximum values for a particular pattern showed significantly lower coefficients for others. This finding shows that our original classification was robust and is now included as Supplementary Fig.3

Photostimulation near a silicon probe can, under certain conditions, induce spike-like artifacts. The authors should address whether the responses with very short latency and low jitter, like those of the rightmost cell in Fig. 1d, might be contaminated by stimulation artifacts. It would be useful to see the raw data (voltage vs time), with spike times highlighted, within a short window of stimulation over several trials for this cell.

R:

It is true that photostimulation can cause undesirable artifacts. However, this is not the case in our data. The anatomical distances between the stimulation region in M1 and the recording region in the motor thalamus are considerably apart, at least 2.6 mm away in the anterior posterior axis and 3 mm in the vertical axis. In our experimental experience, spike-like artifacts with produced by photostimulation are only visible at very short distances of a few dozens of microns. Second, artifacts of electrical, movement or photo origins produce signals with similar amplitudes in all recorded channels. In our recording configuration and spike sorting semi-automatic methods, these kinds of artifacts are

easily discarded, especially in configurations with electrodes that are anatomically close to each other. Our recordings were performed with a silicon probe with a tetrode configuration (25 μm between electrodes). To clarify the reviewer's concern, in the following figure we display the raw spike shapes in the four channels of the tetrode (panel A) and a portion of the raw traces aligned to one optic stimulus (panel B) for the disputed unit (rightmost cell in our original Fig. 1d). Furthermore, in panel C, we display the raster for this neuron in two temporal scales: the one reported in Fig. 1D in the upper panel, and a close up to 300 ms around the first stimulus of the optical trains in the bottom panel. It can be observed that the spikes actually have a nice jitter after the offset of the stimulation.

Figure R4. **A)** Spike traces recorded in the four the tetrode-like configuration channels for the neuron depicted in far-right panel in Figure 1d. **B)** Raw traces (150 ms; upper panel) for the same four channels depicted in A aligned to the onset of a photogenic light stimulus. Spike waves from the neuron in A are highlighted in blue. The lower panel corresponds to the same signals but after high-pass filtering. Blue vertical lines in the bottom of both panel represent detected spikes. **C)** Raster plots for the same neuron. In the upper panel, the same raster as the one depicted in Figure 1d (rightmost panel; -0.5 to 2 seconds around the first stimulus of the rain). For better appreciation of the spike jitter after stimulation, the lower panel, depicts 300 ms of the spiking activity around the first stimulus of the train. It can be seen that the first, short-latency component of the response also presents jittering.

SNr stimulation has a robust inhibitory effect on the example neuron in Fig. 7b, but the effect on the population in the rest of the figure appears to be quite small.

R:

As described in the main text, we found a statistically significant differences in the amplitude of the short latency components of the population response between SNr

paired stimulated and non stimulated conditions, specially for stimuli 3 to 5 of the optic trains. To further clarify these differences and thanks to the reviewer's comment and another observation from Reviewer #2, we are now including the statistical indications in original Fig. 7e (now Fig. 5e).

There are a few typos in the figure legends (e.g., "Craneotomy" and "probe's trackt" in Fig. 1a, c).

R:

We have re-revised the whole manuscript paying special attention to correct this kind of mistakes.

Reviewer #4 (Remarks to the Author):

Review

The paper describes a research study on the responses of the motor thalamus to M1 stimulation in mice, using optogenetics, behavioral tasks, and various analytical methods, exploring the organization and functional significance of thalamic activity patterns. The paper focuses on the effects generated by two stimulation frequencies, and the potential role of these thalamic patterns in motor control and behavior.

By recording from the same neurons during both passive M1 stimulation and active movement execution, the researchers could directly compare the response patterns elicited in each context. They used principal component analysis (PCA) and silhouette-based clustering to categorize neurons into groups based on their response patterns, revealing correlations between the stereotypical patterns evoked by M1 stimulation and the diverse activity patterns related to movement execution. The study found that specific stereotypical response patterns elicited by M1 stimulation were directly linked to distinct phases of forelimb movements during the behavioral task. This linkage was evidenced by comparing the prevalence of response patterns during M1 stimulation with those observed during movement, showing that certain thalamic patterns are preferentially associated with specific movement-related activities.

This research is a tour-de-force, whose major message, in my view, is that the reductionist approach can work for thalamocortical functions in motor control. That is, studies like this, linking specific motor components to specific thalamocortical patterns, allow detailed systematic studies on behavior-relevant circuits with non-behaving and anesthetized animals. This empowers future research by expanding the scope of questions that can be asked, and tools that can be applied, in future studies.

My concerns are moderate, and do not reduce my enthusiasm from this impressive work.

R:

We thank the reviewer for the encouraging comments and for the time spent to revise our work.

Major

1. Previous studies on thalamocortical dynamics typically used more than two frequencies, showing that the frequency is an important factor. The authors thus should address this issue, explaining why they chose these specific frequencies, justify their choice in the context of motor control in this specific system and then discuss the limitations this choice induces.

R:

We chose the range of 300 - 500 ms ISI based on previous reports indicating these frequencies as a proxy of the speed of forelimb movements in rodents when walking or trotting (Rueda-Orozco et al, 2015; Hidalgo-Balbuena et al, 2019) and this temporal range is consistent with multiple goal directed behaviors, such as reaching movements like the ones reported here (Pimentel-Farfan et al, 2022; Park et al, 2022) or even fast sequences of movements (Wolff et al, 2022). These frequency ranges have been also used in previous reports *in vitro* conditions (Goel & Buonomano, 2016) and in anesthetized animals (Hidalgo-Balbuena et al, 2019; Peña-Rangel et al, 2021) to study basic population dynamics in cortical and subcortical networks. However, this important concern was also expressed by Reviewer #2. In our response, we are expressing that in early experiments we also performed a 100 ms ISI protocol (please refer to Fig. R1 and response 3 to Reviewer #2 on this rebuttal file), where the frequency of stimulation interrupted the natural course of the stereotypical responses.

The limitations of our approach are also related to the technical fact that our stimulation is impacted on all cortical projections at the same time, including those projecting and not projecting to the thalamus. We are also limited by the lack of specific knowledge of the natural schemes of communication between the cortex and the thalamus during the execution of behavior. In future experiments we will address this issue by recording cortical and thalamic activity simultaneously.

We have included these clarifications in the discussion section.

2. The abstract is not clear enough in what are nPAPs, and specifically what is the nature of the principal nPAPs reported here. Please make an effort to better describe these aspects – it will facilitate the understanding of most of the readers.

R:

We have modified the abstract including a succinct definition for nPAPs.

Minor

1. In Fig. 1f, please describe what is shown exactly. A single cell? Population? What does the graph represent?

R:

We are depicting the average population response to the first stimulus of the train. This is now clarified in the figure legend.

2. I suggest to address also the dynamics shown during the 5-stimuli trains. Previous studies (also involving thalamocortical circuits) showed that a stabilization process for response latency, in which longer latencies are associated with higher frequencies, such as shown in Fig. 1, might indicate an active closed-loop dynamics (phase locked loop like). Although secondary to the aims of this research, such insights might be important for understanding thalamocortical dynamics during motor control.

3. Thalamic responses to passive stimulations show a kind of a slow wave superimposed on, or modulating, the faster stim-by-stim responses (e.g., Fig. 1j). This dual dynamics deserves a discussion and reference to previous relevant studies (e.g., Yu, Horev Cerebral Cortex 2015). I also suggest to test whether this slow dynamics is linked somehow (specific cells maybe) to the slow dynamics shown by thalamic neurons during movement execution (Fig. 5).

R:

We thank the reviewer for these interesting observations we had not noticed before. The existence of a slow component in the responses we see is possible. If this is confirmed, it is also possible that it may represent a dual dynamic that may expand from M1 stimulation to behavioral control. It would be necessary to determine if this component reflects the spiking activity of individual groups or if it would be present in all neurons. It would be also necessary to determine if it is the result of slowly adapting dynamics to the progression of the stimulating train, or if it would be a component triggered by the first stimulus of the train. Furthermore, analyzing the possibility of closed-loop dynamics embedded in our data is also a very interesting possibility, however, starting to explore these alternatives would deserve a complete effort that may implicate new experiments and that definitively goes beyond the scope of the current work. However, we are including a specific comment in the discussion section and the suggested reference.

Point-by-point response

Preconfigured cortico-thalamic neural dynamics constrain movement-associated thalamic activity

Perla González-Pereyra, Oswaldo Sánchez-Lobato, Mario G. Martínez-Montalvo, Diana I. Ortega-Romero, Claudia I. Pérez-Díaz, Hugo Merchant, Luis A. Tellez, Pavel E. Rueda-Orozco*.

REVIEWER COMMENTS

Reviewer #1 (Remarks to the Author):

In the response letter authors say that they did requested analyses, however I do not see it in the manuscript. The main role of a reviewer is to help to improve manuscript, thus describing results only in the letter is not a proper revision.

As I wrote before in point #3: “The main weakness of this manuscript is that the authors only use one type of stimuli and one type of behavior. “ I suggested to alleviate this problem by additional analyses to compare nPAPs to spontaneous activity. Authors agreed on value of those analyses by writing in the letter: “an approach like the one proposed by the reviewer would significantly strengthen our interpretations.” However, despite describing results of those analyses in the letter, authors did not show it in the manuscript. In the main text authors mention “spontaneous activity” but it is unclear what they exactly did there. There is no information how “spontaneous activity” is defined, and plots 4 i&j, supposedly presenting analyses of spontaneous activity are showing something completely different: relation between activity evoked by M1 stimulation and behavior (at least this is what labels and legends describe).

Considering that current figure 4 is already crowded, maybe making a new figure showing sample activity during spontaneous period (e.g. few seconds before stimulation or before behavior) and presenting similar plots to fig 4h-j, where y-axis would be corresponding to spontaneous activity, could improve clarity of those analyses. The description of new analyses should also be significantly improved for clarity. As authors agreed that such analyses significantly strengthen interpretations, it would be also good to add a sentence also in the abstract that nPAPs have similar structure as spontaneous patterns.

My other comments were largely addressed by authors.

We apologize for this, as the reviewer mentions, we agreed to their suggestions and performed the suggested analysis and described our procedures in the rebuttal letter, which we expected to be also published along with the MS. To substantiate this important observation, we are including a complete and detailed description in the results section. Regarding Fig. 4i, the “Y” axis was incorrectly labeled as Corr. coef. mov, instead it should read “Corr. spont. Act.” As it depicts correlation values between pairs of neurons with low or high correlation values during M1-Stimulation. This panel corresponds to what is

described in our response to Reviewer's comment #3 in our rebuttal letter. We deeply apologize for this misunderstanding, this is now corrected in panel "C" of the new Fig. 6, the corresponding figure legend and the main text. Also, following the reviewer's advice, we have now divided our original figure 4 into Fig. 5 and 6, with the later one including representative spontaneous and M1-evoked activity for three simultaneously recorded neurons as well as the suggested scatter plot.

The amendments to the main text reads as follow:

"On the other hand, previous reports in different brain regions have shown that evidence of preconfigured dynamics can be found in spontaneous activity. To explore this possibility, we analyzed if pairs of neurons that were simultaneously recorded and that presented high correlations during M1 stimulation (Fig. 6a upper panel, higher correlations than the 99.5 percentile of surrogated activity; see methods) would also present higher levels of correlation during spontaneous activity. Spontaneous activity was obtained in the same sessions than the behavioral and optical quantifications, but during periods where no stimulation or behavioral protocol was applied. Under these conditions, we were able to extract 2,145 pairs that were recorded in average 21.6 minutes (range 15.5 to 25.6 min) depending on the length of the session and the number of stimulation protocols and trials performed during the task. Pearson correlation coefficients were calculated from the spontaneous spiking activity of pairs of neurons discretized in 100 ms bins and smoothed with a kernel gaussian filter (200 ms; Fig. 6a lower panel). Then, while the correlation between correlation values during spontaneous activity and M1 stimulation was not significant (Fig. 6b), we found that that the highly correlated pairs during M1 stimulation showed significantly higher correlations values during spontaneous activity than those from low-correlated pairs during M1 stimulation or correlation values obtained from surrogated spike trains (Fig. 6c). To further explore the existence of a latent organization, we wondered if the response sign (activation or inactivation), amplitude and latency to M1 stimulation could predict the same variables during movement execution. To this aim, for each neuron we first calculated the z-score-based signed amplitudes to movement or reward onsets (positive and negative values corresponding to increases and pauses in activity, respectively). Then, we calculated the correlation coefficients between the response amplitudes obtained from the data aligned to movement onset (or reward onset) and the optogenetic stimulation onset and found significant correlations for both (Fig. 6d). After this, to determine a possible temporal relationship between neural activity during M1-stimulation and during movement execution, for each pair of simultaneously recorded neurons we constructed cross-correlation histograms during with the neural activity during movement onsets (from -2 to 2 seconds around movement onsets) and calculated the latency of the center of mass of the each cross-correlogram. Then we calculated the difference in response latency between each pair of neurons during M1 stimulation. At the end, the latencies from both conditions were correlated, rendering a significant relationship between the two variables (Fig. 6e)."

Following the reviewer's advice, we have also modified the abstract included a sentence regarding spontaneous activity as follows:

"... Moreover, they were experience-independent, present in virtually all animals and pairs of highly correlated neurons during M1-stimulation also presented higher correlations during spontaneous activity ..."

We thank the reviewer for their helpful comments and the recognition of our work in addressing the remaining comments.

Reviewer #2 (Remarks to the Author):

Remarks to the Author

The finding that M1 influences the thalamus aligns with previous hypotheses, suggesting its potential impact on lever press behavior is not entirely novel. Notably, the Karel Svoboda group has previously demonstrated a causal contribution of the thalamus in lever pressing. Introducing the concept of nPAPs to elucidate the role of cortico-thalamic circuits in movement might be novel, suggesting a new way to integrate information from the motor cortex and basal ganglia at the thalamus level. Nonetheless, the identification of nPAPs appears to be based solely on observing non-temporally adapting firing in the thalamus influenced by M1, raising the question of whether this is sufficient to assert the existence of nPAPs. As previously pointed out, I have concerns about the lack of enough functional data to support the conclusion of this work.

Reviewer's Comments

1) The authors' current optogenetic experiments lack specificity because they express and stimulate ChR2 in M1 while recording from VL/VM neurons. This approach limits the interpretation of their findings on the role of M1-VL/VM circuit in generating nPAPs. To address this limitation, the authors could inject a canine AAV-loxp-ChR2 virus into VL/VM thalamic nuclei and a CRE virus into M1. This strategy would enable selective M1-VL/VM circuit stimulation using optic fibers or optrodes implanted in the thalamus. Without this refinement, whether the observed nPAPs result solely from M1 input remains uncertain. In addition, the loss function optogenetics using halorhodopsin is necessary to support the input-specific roles in generating nPAPs.

We agree with the reviewer, as we mention in our original response the complete characterization of the specific inputs to the thalamus producing nPAPs is and objective we have in our scope, and that would require much more specific strategies, that we have now been actively implementing on in our lab. We also mentioned that our SNr activation experiments started showing evidence in this direction. However, to further address the reviewers concern, we took advantage of ongoing experiments where we implemented different strategies to specifically target cortico-thalamic subpopulations, including one similar to the one suggested by the reviewer. In brief, we have implemented methods to specifically target pyramidal tract (PT) and corticothalamic (CT) neurons. Our data suggest that the general architecture of M1-VL/VM nPAPs is principally related to CT with a marginal component of PT. The new experiments tackling these projections are displayed in a new main figure (Fig. 3) and supplementary figure (Supplementary Fig. 5) and described in the result section as follows:

“Regarding the origin of these patterns, cortico-thalamic projections to VL/VM arise from two neural subpopulations, PT and CT neurons²³. To disentangle the contribution of PT and CT subpopulations to these potentially preconfigured responses, we implemented the following approaches. For PT neurons (n =5 animals) we implemented an anatomically based strategy with two viruses (Fig. 3a) as the one reported before for IT neurons in rats³⁴. Here, we first induced the expression Cre recombinase (Cre) in M1 cortico-

thalamic projecting neurons (including PT and CT) by injecting a retrograde virus (AAV-retrograde-pgk-Cre AAV) in the VL/VM. Considering that PT neurons (but not CT neurons) projects to both, to the VL/VM and the dorsolateral striatum (DLS), four weeks after the first injection we induced the Cre-dependent expression of ChR2 and GFP in PT neurons by performing a second injection of a retrograde virus (in the ipsilateral DLS to the first injection; pAAV-EF1a-doublefloxed-hChR2-eYFP; Fig.3a). Four weeks after these injections, we observed unilateral rYFP expression in layer V of M1/M2 regions (Fig.3b) and we were able to optically evoke PT activity in M1 (Supplementary Fig. 5a-e). Then, M1 stimulation with the 300ms ISI protocol induced weak thalamic responses characterized mainly by discrete pauses in activity induced by each stimulus of the train (Fig. 3c). This manipulation did not induce strong short-latency increases or long-latency rebounds observed with the original strategy, suggesting a marginal contribution of PT subpopulation to the described population dynamics. To further confirm this possibility, we performed the same 300ms ISI protocol but using 50 ms light pulses (as opposed to the 5 ms ones used in previous manipulations) and with maximum light power. This manipulation induced similar dynamics to the 5 ms based protocol but with slightly stronger pauses but importantly, the short- and long-latency increases were also almost absent (Supplementary Fig. 5f).

To investigate CT and PT projections, we retrogradely induced the expression of ChR2 in PT and CT (PT+CT) neurons by injecting a retrograde virus in VL/VM (pAAVretro-Syn-ChR2(H134R)-GFP; n = 2 animals), and four weeks after we performed the same stimulation protocol producing very similar population responses to those observed in the original unspecific strategy depicted in Fig. 2a (Fig. 3d-f). Then, to further isolate the CT component, in a new group of animals (n = 3) we attempted to express ChR2 in CT neurons and eliminate PT, IT and CC neurons in the ipsilateral M1 to the recorded VL/VM. To this aim we introduced the following strategy: in a first surgery we performed three injections, a retrograde virus in the VL/VM to express ChR2 and GFP, and a retrograde virus in the ipsilateral DLS and the contralateral M1 to induce the expression of Cre (same virus as the ones used in the previous experiments). In this way, both, PT and CT neurons would express ChR2, and PT, IT and CC (but not CT) neurons would express Cre in the ipsilateral side to the recorded VL/VM (Fig. 3g). Four weeks later, a single injection of the virus pAAV-flex-taCasp-TEVp in the ipsilateral M1 to the recordings site induces the Cre-dependent expression of caspase 3 in PT, CC and IT neurons ultimately resulting in their apoptosis (Fig. 3g). Four weeks after the last injection we observed ipsilateral expression of GFP and a decreased signal in layer V (Fig. 3h). Here again the M1 300 ms ISI stimulation protocol induced similar population responses to those observed in our previous experiment (PT+CT) and our original observations (Fig. 3i). Finally, we explored the existence of nPAPs evoked by specific subpopulations. For this analysis, we pooled all neurons recorded from all groups, that is, the general M1 stimulation group displayed in Fig. 2, and the PT, PT+CT and CT groups, and re-applied our PCA/Silhouette-based method to identify groups of neurons with similar response patterns and compared their occurrence proportions. We found that in the isolated PT group the patterns with short-latency and long-latency activation were significantly reduced, while both groups PT-CT and CT presented all five patterns of activation (Fig. 3j) with similar proportions than the original group (Fig. 3k), except that PT+CT group expressed a significantly higher proportion of pattern 4 than the original M1 group. Altogether, these data suggest that M1-evoked nPAPs are mainly related to CT projections with a marginal component of PT neurons.”

Altogether, this data suggests that VL/VM nPAPs reported here are mainly related to CT neurons projections with a marginal component of PT neurons. On the other hand, we believe that the previous experiments and manipulations attest better for the specific contribution of the different cortico-thalamic inputs than halorhodopsin-based strategies. Furthermore, as the reviewer mentioned, previous reports already indicate that inhibiting the motor thalamus would result in a generalized decrease in activity of the motor cortex and vice versa (Guo et al, 2017; already cited in the original version of the MS).

-In the optogenetic experiment of SNR, they only have n=4 mice (SNR stimulation n=2 / SNR-VL/VM stimulation, n=2).

They combined two groups to conclude. Additional experiments seem necessary to clarify the role of SNR.

We reasoned that because we found very similar behavioral effects in the four animals, the results could be pooled together. To support this strategy, we are now including the median and 25th and 75th percentile values for each animal and each variable.

2) The authors mentioned, “More importantly, we found that 12 out of the 18 animals exhibited the five patterns, two out of 18 presented four patterns, and only four animals presented only one pattern.” If nPAPs function as information packets, each with behavioral meaning, how can it be explained that some individuals have more (5) or fewer (1) information packets?

We did not conclude that the animals that exhibited fewer patterns presented fewer information packets. We used the concept of information packets as a general reference for our study, information packets are described as coordinated sequential activation involving various neurons, in this context, the modification in the prevalence of a particular pattern would result in the modification of the general architecture of the information packet. For example, in the new Fig. 3, the activation of PT neurons presented patterns characterized by pauses.

On the other hand, while we cannot rule out the possibility of an underlying structural difference in these animals, the fact that we found four to five patterns in the rest of the animals including naïve and trained (Fig. 5b), under anesthesia (Supplementary Fig. 4), and in the isolated CT preparation (Fig. 3), suggest that this effect may be related to variability in our recording conditions, for example, the variability in the final position of the optical fiber or recording probe, or the total amount of neurons recorded on those animals. For example, in the original group depicted on Fig. 1 and 2, two out of the four animals that presented only one pattern also presented less than 15 well isolated units, while in average, we were able to record about 47 cell per animal.

We have included these arguments in the discussion section.

- The number and portion of specific patterns must be described statistically. Otherwise, this numerical description is empirical and dampens any conclusion from this finding.

(a) Does an individual showing only one pattern have four other patterns for five?

(b) Or does it only have one?

What does each case (a and b) mean?

The answer is b, individuals showing only one pattern did not show any other pattern. We have now included this clarification in the main text.

On the other hand, the proportions of specific patterns are statistically described in boxplots constructed from sessions where more than 10 cells were recorded simultaneously. These proportions are displayed in Fig. 2 h (upper panel), Fig. 3 h, right panel (now Fig. 4) Fig. 4b (now Fig. 5) and Fig. 5h (now Fig. 7).

3) Figures 1 and 2 have virtually the same title. If the two figures reach the same conclusion, they should be combined into one figure. On the other hand, if the two figures reach different conclusions, their titles should be distinctly different.

Figures 1 and 2 display different characteristics of the M1-evoked responses. In Fig. 1 we focus on our general approach, the heterogeneity in responses observed in the individual raster plots and the response latencies and the amplitude to the first and the full stimulation train. Figure two is dedicated to population organization and how individual neurons can be classified into specific groups based on their response patterns to M1 stimulation. We have now changed the titles of these figures to better reflect these differences and now reads as follows:

Fig. 1 Prototypical M1-evoked responses in VL/VM

Fig.2 VL/VM responses can be classified into distinct M1-evoked patterns.

4) As the authors mentioned in the main text, why do anesthetized mice still have stable preconfigured response patterns while there are no movements? Even in Supplementary Figure 4, there seems to be no difference in the occurrence and amplitude of response patterns between awake and anesthetized animals. How can it be concluded that nPAPs contain information about movement?

We did not conclude that nPAPs contain movement information, but that nPAPs constrain movement-related dynamics. For example, behaviorally related pattern type 5 consisted of transient decreases in activity followed by a rebound and is statistically related to M1-evoked pattenr type 4.

Minor Comments

In the main text, it is written, “Some patterns presented ramping activity with the highest peaks coinciding with movement onset (Fig. 3e; types 4, 6, and 7); one group presented ramping activity (Fig. 3g; type 2) or different levels of transient inactivation during movement maintenance (Fig. 3g; types 3, 8, and 9)”. Shouldn't it be Fig. 3g and not Fig. 3e?

Reviewer is right, this is now corrected in the new version of the MS.

Many Supplementary Figures still lack titles.

We have added titles in all supplementary figure legends

It is recommended that two brain regions be represented, imaged from one individual, with eYFP expression, as shown in Figure 1c.

We have proceeded according to the reviewer's suggestion.

The peri-event histograms showing the firing rate have varying y-axis labeling (e.g., F. rate, Firing Rate, Rate). Adding a scale bar or unit bar showing firing rate values would also be beneficial.

We have homogenized everything to F. rate, and now we are including the maximum firing rates for each histogram.

Reviewer #3 (Remarks to the Author):

I appreciate the authors' revisions, which have improved the manuscript and addressed several of my concerns. The paper provides valuable information about the effects of M1 stimulation on thalamic activity, and I think this is a strength. However, the central claim of preconfigured activity patterns still relies on a vague definition of these patterns and indirect experimental evidence for their existence, which limit the rigor, conceptual advance, and expected impact of the study. The authors take inspiration from compelling studies of precise spike timing in the hippocampus, where sequential activation of place cells is recapitulated, temporally compressed, and often reversed during sharp wave ripples. The unambiguous demonstration of replay (Foster & Wilson, *Nature* 2006; Diba & Buzsaki, *Nat. Neuro.* 2007) depended critically on (1) a precise operational definition of the phenomenon (the sequence of cells that spike during SWRs should be the same as the sequence during track traversals, or should be the same sequence in reverse), (2) visualization of clear examples of individual replay events, and (3) an analysis of the distribution of a statistic computed within individual events (in the cited studies, the rank correlation between spiking order). While subtle statistical challenges remain (see Tingley & Peyrache, *Proc. Royal Soc. B*, 2020), the existence of SWR replay has been clearly established, and has had a major impact in the hippocampal field and beyond.

In my opinion, the current study does not meet these three criteria. I am still unable to determine what kinds of patterns are and are not preconfigured, and whether the former can be identified in these data. The authors should state clearly what they mean by preconfigured patterns - is a preconfigured pattern a specific sequence of neural firing, which occurs both during natural behavior and in response to stimulation of an input? If so, analyses should be focused on testing for a concordance between firing sequences in the two situations. The closest analysis in the current manuscript is in Fig. 4j, which shows a correlation between the time of cross-correlation peaks for pairs of cells during stimulation and reaching. This correlation is relatively weak, and a null distribution does not seem to be computed (though it is for the amplitudes in 4h). More importantly, this is an indirect way of determining the similarity of temporally-extended stimulation- and reach-aligned firing sequences. If the authors show heatmaps of reach-aligned activity, sorted by the stimulation-aligned firing order, is a diagonal structure present? Additional analysis methods, such as those previously used to identify hippocampal replay, or those

used to examine low-dimensional cortical dynamics in primate reaching studies, could enable the authors to really focus on the central issue of how similar the stimulation- and reach-related patterns are.

We thank the reviewer for the time spent in the revision of our MS. As a general conceptual definition, nPAPs have been described as organized activity parcellated into groups of neurons. This general definition is already stated in the abstract of our manuscript. But in the context of this work, involving inter-structure communication between M1 and VL/VM, an operational definition could be expressed as “*highly reproducible response patterns expressed by groups of individual neurons*”. The activity patterns at the single cell level are described in the introduction of our manuscript as follows:

“... in the rodent cortical and subcortical areas, including input and output nuclei of the basal ganglia (BG), cutaneous whisker or forepaw stimulations induce diverse activity patterns at the single cell level, generally characterized by a short-latency response followed by a transitory inactivation period, and usually a second, long-latency response⁸⁻¹².”

In the introduction we also stated that these basic *single cell level activity patterns* appear to compose more complex population activity patterns, such as sequences of activation. This is stated as follows:

“In turn, these complex population activity patterns appear to be composed of various activity patterns at the single neuron level.”

After the reviewer’s comment, we have now included these arguments and definition in the first paragraph of our discussion.

Is important to clarify that while we took inspiration from different groups working on cortical and hippocampal dynamics, typically associated to cognitive function, this was not the only reference framework we used. We also referred to previous work on population dynamics in cortical and striatal motor regions, and is important to highlight that in our opinion, is still difficult to bridge the cognitive and motor domains. By using this framework, we were able to compare some basic features of the previously reported literature, for example, sequential organization, commonly observed in both “hippocampal-related” investigations, such as the examples provided by the reviewer, but also motor networks, such as the examples cited in our MS. Is also important to clarify that it was not our intention to make a complete extrapolation of the hippocampal dynamics to motor-related related dynamics, in this case, thalamic dynamics. For example, recapitulation or reverse activation has not been reported during sleep in motor regions such as the striatum or the motor thalamus.

Regarding the contribution of single cell level activity patterns to population sequential activation, we observed sequential activation triggered by each M1 stimulus on both, 300 and 500 ms ISI protocols (Fig. 2). However, under these conditions, we did not observe signs of adaptation to the different stimulation protocols (Fig. 2 and Supplementary Fig.

1). These fixed dynamics differed from what was previously reported for striatal and cortical regions, in awake, anesthetized, and even in ex-vivo preparations, that is, expansion or contraction of the neural sequences depending on experimental conditions (Mello et al, 2015; Monteiro et al, 2023; Goel & Buonomano, 2016). This now highlighted in the discussion section. Hence, we focused on the activity patterns at the single cell level and found that cells could be grouped into clusters with similar response patterns to M1 stimulation or in relationship with movement execution. On the other hand, as can be observed in the individual examples and population activity of Fig. 4, behaviorally related thalamic activity was not obviously associated to sequential activation, but more to sustained activations/inactivations during movement maintenance. For these reasons, we were not expecting to see a translocation of the sequential “diagonal” activity from M1-evoked activity to behaviorally related activity. However, the most obvious relationship that we observed between M1-evoked activity and behaviorally related activity is depicted in Fig. 4(now 6). That is, neurons that increased or decreased their activity during movement execution (Fig. 4c), expressed higher or lower amplitudes in response to M1-stimulation (Fig. 4d), respectively. They also expressed higher levels of adaptation to the progression of the stimulation train (Fig. 4e), and in the case of M1-evoked pattern type 4, was significantly associated with behaviorally triggered pattern type 5 (Supplementary Fig. 8).

Regarding clear examples of the activity patterns, these are depicted in the raster plots of individual neurons (Fig. 1d, 2g, Supplementary Fig. 4d) and in the averaged activity of groups of neurons classified as part of specific response type (Fig. 2f, 3j, 5b, 7g, Supplementary Fig. 4c, 8d). Moreover, in response to one of the original comments of Reviewer #3, we plotted the average responses of all individual neurons classified as all belonging to specific pattern types (Fig. R3 in our original Point-by-point reply).

In summary, while the evidence we present in this MS does not directly translocate to the reply of sequential activation observed in the hippocampus, our data in support of preconfigured dynamics in motor regions includes, robust thalamic M1-evoked patterns expressed by clusters of individual neurons and visible in most of the animals, independent from learning and present in at least two opposite brain states, awake and anesthetized. These dynamics can be partially associated with cortical subpopulations, mainly CT neurons with a marginal contribution of PT neurons. On the other hand, pairs of neurons with higher correlation coefficients during M1-stimulation would also present higher correlation coefficients during spontaneous activity. Finally, the magnitude and sign of M1-responses can partially predict if the neurons would present increased or decreased responses during movement execution, and we could observe that at least one M1-evoked pattern was associated to a behaviorally triggered pattern.

Reviewer #4 (Remarks to the Author):

The authors addressed all my concerns. Congratulations for an excellent scientific contribution.

We thank the reviewer for the time spent in the revision of our manuscript and their encouraging words.

Point-by-point response

Preconfigured cortico-thalamic neural dynamics constrain movement-associated thalamic activity

Perla González-Pereyra¹, Oswaldo Sánchez-Lobato¹, Mario G. Martínez-Montalvo¹, Diana I. Ortega-Romero¹, Claudia I. Pérez-Díaz¹, Hugo Merchant¹, Luis A. Tellez², Pavel E. Rueda-Orozco^{1*}.

¹ Departamento de Neurobiología del Desarrollo y Neurofisiología, Instituto de Neurobiología, UNAM

² Departamento de Neurobiología Conductual y Cognitiva, Instituto de Neurobiología, UNAM

*Correspondence: Dr. Pavel E. Rueda-Orozco; e-mail: pavel.rueda@gmail.com; ruedap@unam.mx;

REVIEWERS' COMMENTS

Reviewer #1 (Remarks to the Author):

Authors largely addressed my concerns.

Minor comments:

1) In sentence "previous reports in different brain regions have shown that evidence of preconfigured dynamics can be found in spontaneous activity." authors should provide references to those papers.

R.

References are now included

2) In the new figure some results are marked as statistically significant with '*', however in main text there is mention what were p values and what test was used. Please provide this information.

R.

The information is now included

Reviewer #2 (Remarks to the Author):

Congratulations. This finding significantly contributes to our understanding of the thalamus's role in integrated sensory-motor information processing. I have no further comments.

R.

We thank the reviewer for their encouraging comments.

Reviewer #3 (Remarks to the Author):

I appreciate the authors' revisions, which have further improved the manuscript. However, while it provides valuable data about the effects of motor cortical stimulation on thalamic spiking in the mouse, I remain unconvinced that these data exhibit a novel phenomenon of preconfigured patterns in the motor system. The definition of such patterns as "highly reproducible response patterns expressed by groups of individual neurons," which includes responses evoked by electrical or optogenetic stimulation, is extremely broad, and could apply to the results of almost any stimulation experiment. Numerous studies over the past sixty years have applied microstimulation to one sensory or motor area while recording single-unit discharges in another, and have shown the temporal properties of these discharges can vary across neurons. This is fully expected based on differences in intrinsic excitability, synaptic strength, conduction delays, and similar considerations.

To put it differently, I am still unable to determine what the null hypothesis was. If the experiments were designed to test for heterogeneity in thalamic responses to cortical stimulation (which the authors interpret as preconfigured patterns), what plausible experimental outcomes would have provided a negative answer?

R.

We thank the reviewer for their comments throughout the revision process.

To directly answer the question, one possibility is that corticothalamic communication would express non rigid M1-evoked patterns, for example, that would change over the course of learning. Another possibility is that their prevalence would be also less rigid and not present in all the animals. Finally, another possibility is that the expression of these patterns would not be linked to specific components of the anatomical architecture of the thalamic inputs.